# Cell-cycle dependent phosphorylation of yeast pericentrin regulates γ-TuSC-mediated microtubule nucleation

Tien-chen Lin[1,2], Annett Neuner[1], Yvonne T Schlosser[1], Annette ND Scharf[1], Lisa Weber[1], Elmar Schiebel[1]*

[1]Zentrum für Molekulare Biologie (ZMBH), Universität Heidelberg, Heidelberg, Germany; [2]The Hartmut Hoffmann-Berling International Graduate School, University of Heidelberg, Heidelberg, Germany

**Abstract** Budding yeast Spc110, a member of γ-tubulin complex receptor family (γ-TuCR), recruits γ-tubulin complexes to microtubule (MT) organizing centers (MTOCs). Biochemical studies suggest that Spc110 facilitates higher-order γ-tubulin complex assembly (*Kollman et al., 2010*). Nevertheless the molecular basis for this activity and the regulation are unclear. Here we show that Spc110 phosphorylated by Mps1 and Cdk1 activates γ-TuSC oligomerization and MT nucleation in a cell cycle dependent manner. Interaction between the N-terminus of the γ-TuSC subunit Spc98 and Spc110 is important for this activity. Besides the conserved CM1 motif in γ-TuCRs (*Sawin et al., 2004*), a second motif that we named Spc110/Pcp1 motif (SPM) is also important for MT nucleation. The activating Mps1 and Cdk1 sites lie between SPM and CM1 motifs. Most organisms have both SPM-CM1 (Spc110/Pcp1/PCNT) and CM1-only (Spc72/Mto1/Cnn/CDK5RAP2/myomegalin) types of γ-TuCRs. The two types of γ-TuCRs contain distinct but conserved C-terminal MTOC targeting domains.

*For correspondence: e.schiebel@zmbh.uni-heidelberg.de

**Competing interests:** The authors declare that no competing interests exist.

**Reviewing editor**: Jon Pines, The Gurdon Institute, United Kingdom

## Introduction

The budding yeast spindle consists of ~40 microtubules (MTs) that extend between the two opposed spindle pole bodies (SPBs). Because of the closed mitosis in yeast, the SPBs remain embedded in the nuclear membrane throughout mitosis. Cell-cycle regulated spindle assembly begins in S-phase with nucleation of MTs onto the surface of the newly assembled SPB. As soon as MTs emanate from both SPBs, they interdigitate (pole to pole MTs) or attach to the kinetochores (pole to kinetochore MTs) at the end of S phase, forming the bipolar spindle (*O'Toole et al., 1999*).

The γ-tubulin complex is the core player in MT nucleation. In budding yeast *Saccharomyces cerevisiae*, two molecules of γ-tubulin (Tub4) assemble together with one molecule of Spc97 (ortholog of human GCP2) and Spc98 (ortholog of human GCP3) into a tetrameric γ-tubulin small complex (γ-TuSC), which is conserved in all eukaryotes (*Geissler et al., 1996*; *Knop et al., 1997*; *Guillet et al., 2011*). The purified γ-TuSC of budding yeast is able to self-oligomerize into symmetric ring-like structures under low salt buffer conditions. The diameter and the pitch of the γ-TuSC ring resemble that of MT cylinder, suggesting that the γ-TuSC ring functions as a MT nucleation template. However, the in vitro nucleation activity of the γ-TuSC assemblies remained poor, presumably because of the suboptimal spacing of every second Tub4 within the γ-TuSC ring that blocks interaction with tubulin in the MT cylinder (*Kollman et al., 2008*, *2010*; *Choy et al., 2009*). The concept of γ-TuSC oligomerization is further supported by in vivo measurements of budding yeast γ-tubulin complex components on detached single MT nucleation sites. The γ-tubulin:Spc97:Spc98 ratio was 2.4:1.0:1.3 with a total of ~17 γ-tubulin molecules per nucleation site (*Erlemann et al., 2012*), suggesting a slight excess of γ-tubulin and Spc98 molecules over Spc97 in the assembled MT nucleation site.

**eLife digest** Microtubules are hollow structures made of proteins that have a central role in cell division and a variety of other important processes within cells. For a cell to divide successfully, the chromosomes containing the genetic information of the cell must be duplicated and then separated so that one copy of each chromosome ends up in each daughter cell. To separate the chromosomes, microtubules extend out from two structures called spindle pole bodies, which are found at either end of the cell, and pull one copy of each chromosome to opposite sides of the cell.

Although the individual proteins that make up a microtubule can self-assemble into tubes, this occurs very slowly, so cells employ other molecules to speed up this process. In yeast cells, a protein called gamma-tubulin is recruited to the spindle pole body by the protein Spc110, where it combines with two other proteins to form a complex called the gamma-tubulin small complex. Several of these complexes then join together to form a ring, which probably acts as the platform that microtubules grow from. Recent observations suggested that Spc110 may help to construct this ring, but without revealing how.

Now, Lin et al. reveal that Spc110 can regulate microtubule formation by controlling how several gamma-tubulin small complexes bind together, and have identified the exact section of Spc110 that interacts with the complexes. However, the Spc110 must become active before it can perform this role, and it is only activated during certain stages of cell division, through phosphorylation. The structures in Spc110 that bind to the gamma-tubulin small complex in yeast are also found in gamma-tubulin binding receptor proteins in human cells. The work of Lin et al. demonstrates that proteins that are assumed to have passive roles within cells, such as Spc110, often play more active roles instead.

In most eukaryotic organisms, such as fission yeast, *Drosophila*, *Xenopus*, and human, multiple γ-TuSCs further assemble with additional GCP family members (GCP4-6) into the larger γ-tubulin ring complex (γ-TuRC) (**Gunawardane et al., 2000**; **Zhang et al., 2000**; **Murphy et al., 2001**; **Anders et al., 2006**). However, these additional GCP proteins are not encoded in the budding yeast genome.

Various proteins are involved in the recruitment of γ-tubulin complexes to microtubule organizing centers (MTOCs) such as centrosomes and SPBs. The small protein Mozart1 is encoded in most eukaryotic genomes except the one of budding yeast (**Teixido-Travesa et al., 2012**). In *Schizosaccharomyces pombe* and *Arabidopsis thaliana* Mozart1 interacts with the GCP3 subunit of γ-tubulin complexes (**Janski et al., 2012**; **Nakamura et al., 2012**; **Batzenschlager et al., 2013**; **Dhani et al., 2013**; **Masuda et al., 2013**). In *S. pombe* Mozart1 appears important for the γ-TuSC recruitment to SPBs (**Dhani et al., 2013**; **Masuda et al., 2013**).

Besides Mozart1, a group of conserved proteins called γ-tubulin complex receptors (γ-TuCRs) are required for targeting γ-tubulin complexes to MTOCs. Most of them carry a highly conserved centrosomin motif 1 (CM1) that interacts with GCP subunits of γ-tubulin complexes (**Sawin et al., 2004**). How Mozart1 and γ-TuCRs cooperate is not understood. However, in budding yeast cells that lack a Mozart1 gene, γ-TuCRs are the sole factors responsible for γ-TuSC recruitment to SPBs. Spc110 is the budding yeast homolog of pericentrin (PCNT) and functions as γ-TuCRs at the nuclear side of the SPB (**Knop and Schiebel, 1997**; **Sundberg and Davis, 1997**). The N-terminal Spc110 encompasses the CM1 that interacts with the Spc98 subunit of γ-TuSC (**Knop and Schiebel, 1997**; **Nguyen et al., 1998**; **Vinh et al., 2002**; **Sawin et al., 2004**; **Zhang and Megraw, 2007**; **Fong et al., 2008**; **Samejima et al., 2008**). In addition, the N-terminal region of Spc110 is phosphorylated in a cell-cycle dependent manner. Phospho-Spc110 appears as cells progress from S phase, continues accumulating during mitosis, and vanishes at the anaphase onset (**Friedman et al., 1996**; **Stirling and Stark, 1996**). Spc110 phosphorylation accounts for the impact of Cdk1 and Mps1 kinases on spindle dynamics (**Friedman et al., 2001**; **Huisman et al., 2007**; **Liang et al., 2013**). However, a clear understanding behind this observation is lacking. Interestingly, when γ-TuSC and an N-terminal fragment of Spc110 (amino acids 1–220 of Spc110; $Spc110^{1-220}$) were co-expressed in insect cells, a filament-like γ-TuSC-$Spc110^{1-220}$ complex formed. The nucleation capacity of this purified γ-TuSC-$Spc110^{1-220}$ complex exceeded that of the γ-TuSC alone (**Kollman et al., 2010**). Thus, $Spc110^{1-220}$ influences γ-TuSC properties with yet unclear mechanism.

Here we have tested the possibility that phosphorylation of the γ-TuCR Spc110 regulates MT nucleation by inducing γ-TuSC oligomerization. Single particle analysis of γ-TuSC incubated with phosphomimetic Spc110 mutant proteins showed that Mps1 and Cdk1 promoted MT nucleation through Spc110 phosphorylation. Phosphorylated Spc110 and the interaction with the N-terminal domain of Spc98 induce γ-TuSC oligomerization. In addition, bioinformatic analysis revealed a conserved region around T18, that we named Spc110/Pcp1 motif (SPM). SPM and CM1 motifs are both important for γ-TuSC binding and oligomerization. A comparison of γ-TuCRs for the presence of SPM and CM1 identified SPM-CM1 (Spc110, Pcp1, PCNT) and CM1-only types of γ-TuCRs (Spc72, Mto1, Cnn, CDK5RAP2, myomegalin) in most organisms. While the SPM-CM1 type of γ-TuCRs carries the PACT domain and is targeted only to the centrosome or the nuclear side of the SPB, the CM1-only type of γ-TuCRs has either a MASC (Mto1 and Spc72 C-terminus) (*Samejima et al., 2010*) or a CM2 motif and is recruited to, centrosomes, the cytoplasmic side of the SPB or acentrosomal MTOCs.

## Results

### Phosphorylation of N-Spc110 at Mps1 and Cdk1 sites is required for efficient interaction with γ-TuSC

To test whether Spc110$^{1–220}$ phosphorylation promoted γ-TuSC ring formation, we purified GST-Spc110$^{1–220}$ (named Spc110$^{1–220}$) from both *E. coli* and the baculovirus expression system. Spc110$^{1–220}$ encompasses Cdk1 and Mps1 phosphorylation sites and the conserved CM1 motif (*Figure 1A*). Because of the post-translational modification machinery, Spc110$^{1–220}$ purified from insect cells harboured phosphorylations on S60/T68 and S36/S91 (*Figure 1—figure supplement 1A–D*), that correspond to established Mps1 and Cdk1 sites, respectively (*Figure 1A*; *Friedman et al., 2001*; *Huisman et al., 2007*). In contrast, Spc110$^{1–220}$ was not phosphorylated when purified from *E. coli*.

We incubated purified γ-TuSC with either the phosphorylated Spc110$^{1–220}$ from insect cells (Spc110$^{1–220-P}$) or the non-phosphorylated Spc110$^{1–220}$ from *E. coli* and then analyzed the protein complexes by gel filtration. Spc110 dimerizes via the coiled-coli region in the centre of the protein (*Kilmartin et al., 1993*; *Muller et al., 2005*). To substitute for the lack of this region, we performed the assays with GST-Spc110$^{1–220}$ that dimerizes via GST–GST interactions. We used TB150 buffer in our assay instead of the BRB80 buffer used by *Kollman et al. (2010)*. BRB80 induces oligomerization of γ-TuSC without the need for addition of Spc110 (*Figure 1—figure supplement 2A,B*). In contrast, in TB150 buffer the majority of γ-TuSC was monomeric (*Figure 1—figure supplement 2C*). TB150 therefore allowed us to monitor the impact of Spc110$^{1–220}$ on γ-TuSC oligomerization by gel filtration. Addition of Spc110$^{1–220-P}$ shifted γ-TuSC into fractions that eluted earlier than when γ-TuSC was run on the columns on its own (*Figure 1B*, shift from fraction 10 to fraction 8). The peak at the void-volume (fraction 8) likely represented γ-TuSC oligomers. In contrast, non-phosphorylated Spc110$^{1–220}$ expressed in *E. coli* did not change the γ-TuSC elution profile. The γ-TuSC pool eluted as monomeric γ-TuSC (*Figure 1B*, fractions 10–11). This result suggests that Mps1 and Cdk1 kinases regulate the interaction between Spc110 and γ-TuSC through phosphorylation of Spc110 N-terminal domain.

### Phosphorylation of N-Spc110 regulates the γ-TuSC oligomerization promoting activity

To further confirm that phosphorylation of Spc110$^{1–220}$ alters the interaction with γ-TuSC, phosphomimetic and non-phosphorylatable Spc110$^{1–220}$ mutant proteins were purified from *E. coli* (*Figure 2—figure supplement 1A*). *spc110$^{2A}$* (Cdk1 sites: S36A, S91A) and *spc110$^{3A}$* (Mps1 sites: S60A, T64A, T68A) have been studied in vivo before and shown to behave as non-phosphorylatable mutants (*Friedman et al., 2001*; *Huisman et al., 2007*; *Liang et al., 2013*). In addition, we analyzed the phosphomimetic Spc110$^{1–220-2D}$ (Cdk1 sites: S36D and S91D), Spc110$^{1–220-3D}$ (Mps1 sites: S60D, T64D, T68D), and Spc110$^{1–220-5D}$ (Cdk1 and Mps1 sites: S36D, S60D, T64D, T68D, and S91D). Migration of all purified Spc110$^{1–220}$ mutant proteins was comparable upon gel filtration, indicating that the mutations did not alter the overall structure of the protein (*Figure 2—figure supplement 1B,C*).

We used gel filtration chromatography to address whether Spc110$^{1–220}$ mutant proteins induce higher-order γ-TuSC oligomers. The phosphomimetic Spc110$^{1–220-2D}$ (Cdk1 sites), Spc110$^{1–220-3D}$ (Mps1 sites), and Spc110$^{1–220-5D}$ (Cdk1 and Mps1 sites) induced an apparent shift of γ-TuSC towards the void-volume compared to the γ-TuSC only run or γ-TuSC plus non-phosphorylated Spc110$^{1–220-WT}$ from *E. coli* (*Figure 2A–C*, *Figure 2—figure supplement 2A,B*). The behavior of the phosphomimetic Spc110

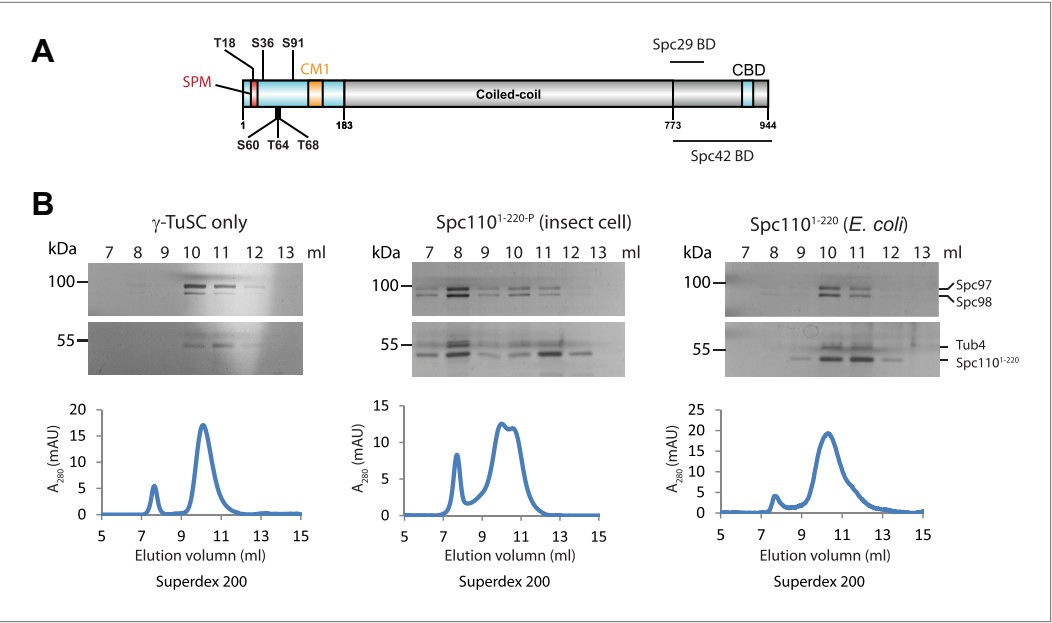

**Figure 1**. Phosphorylation of N-Spc110 is required for the γ-TuSC oligomerization. (**A**) Diagram of Spc110's functional domain organization and the position of phospho-sites investigated in this study. These sites are located at the N-terminal domain of Spc110, which directly interacts with γ-TuSC (**Vinh et al., 2002**). The conserved centrosomin motif 1 (CM1) (**Sawin et al., 2004**; **Zhang and Megraw, 2007**) is also within the N-terminal domain. The C-terminal domain of Spc110 is involved in the interaction with SPB central plaque components Spc29 and Spc42 (**Adams and Kilmartin, 1999**; **Elliott et al., 1999**). CBD: calmodulin binding domain. (**B**) Post-translational modifications of Spc110$^{1-220}$ are required for promoting γ-TuSC oligomerization. Spc110$^{1-220}$ was expressed and purified from insect cells (middle panel) or from *E. coli* (right panel). Only Spc110$^{1-220}$ from insect cells carried post-translational modifications (**Figure 1—figure supplement 1**). Recombinant γ-TuSC was incubated with Spc110$^{1-220}$ or TB150 buffer only on ice. Oligomerization of γ-TuSC-Spc110$^{1-220}$ was tested by gel filtration chromatography using a Superdex 200 10/300 column. Peak fractions of the chromatograms were analysed by SDS-PAGE and silver staining.

The following figure supplements are available for figure 1:

**Figure supplement 1**. Phosphorylation of Spc110$^{1-220}$ from insect cells.

**Figure supplement 2**. γ-TuSC does not oligomerize in TB150 buffer.

mutant proteins reflected that of in vivo phosphorylated Spc110$^{1-220-P}$ purified from insect cells (**Figure 1B**), indicating that the mutations to acidic residues did mimic phosphorylation. In contrast, the γ-TuSC shift was not seen upon inclusion of non-phosphorylatable Spc110$^{1-220-5A}$ (Cdk1 and Mps1 sites) with the γ-TuSC (**Figure 2—figure supplement 2A,B**). Thus, Spc110$^{1-220-5A}$ behaved as the non-phosphorylated Spc110$^{1-220}$ purified from *E. coli*, suggesting that the non-phosphorylatable mutations did not impair the protein. This conclusion was further supported by the observation that Spc110$^{1-220-2D}$ or Spc110$^{1-220-3D}$ still possessed oligomerization-promoting activity when combined with non-phosphorylatable mutations (Spc110$^{1-220-2D3A}$ and Spc110$^{1-220-2A3D}$, **Figure 2—figure supplement 3**). Taken together, the gel filtration experiment suggests that phosphorylated Spc110 induces oligomerization of γ-TuSC.

Because the upper limit of the separation range of the Superose 6 gel filtration column is ~5000 kDa, the γ-TuSC oligomers in the void-volume contain at least 14 copies of γ-TuSC (14-mers). This indicates the presence of ring-like complexes or even higher-order oligomeric structures in the void-volume. To confirm the existence of structurally organized oligomers, fractions corresponding to the void-volume peak (**Figure 2—figure supplement 2A**, fraction 7) were subjected to EM analysis (**Figure 2D,E**, **Figure 2—figure supplement 4A**). Consistent with the low A$_{280}$ absorbance, hardly any oligomeric γ-TuSC species were found in the void fraction of γ-TuSC only or γ-TuSC incubated with the non-phosphorylated Spc110$^{1-220-WT}$ (**Figure 2E**, **Figure 2—figure supplement 4B**). In contrast, oligomeric

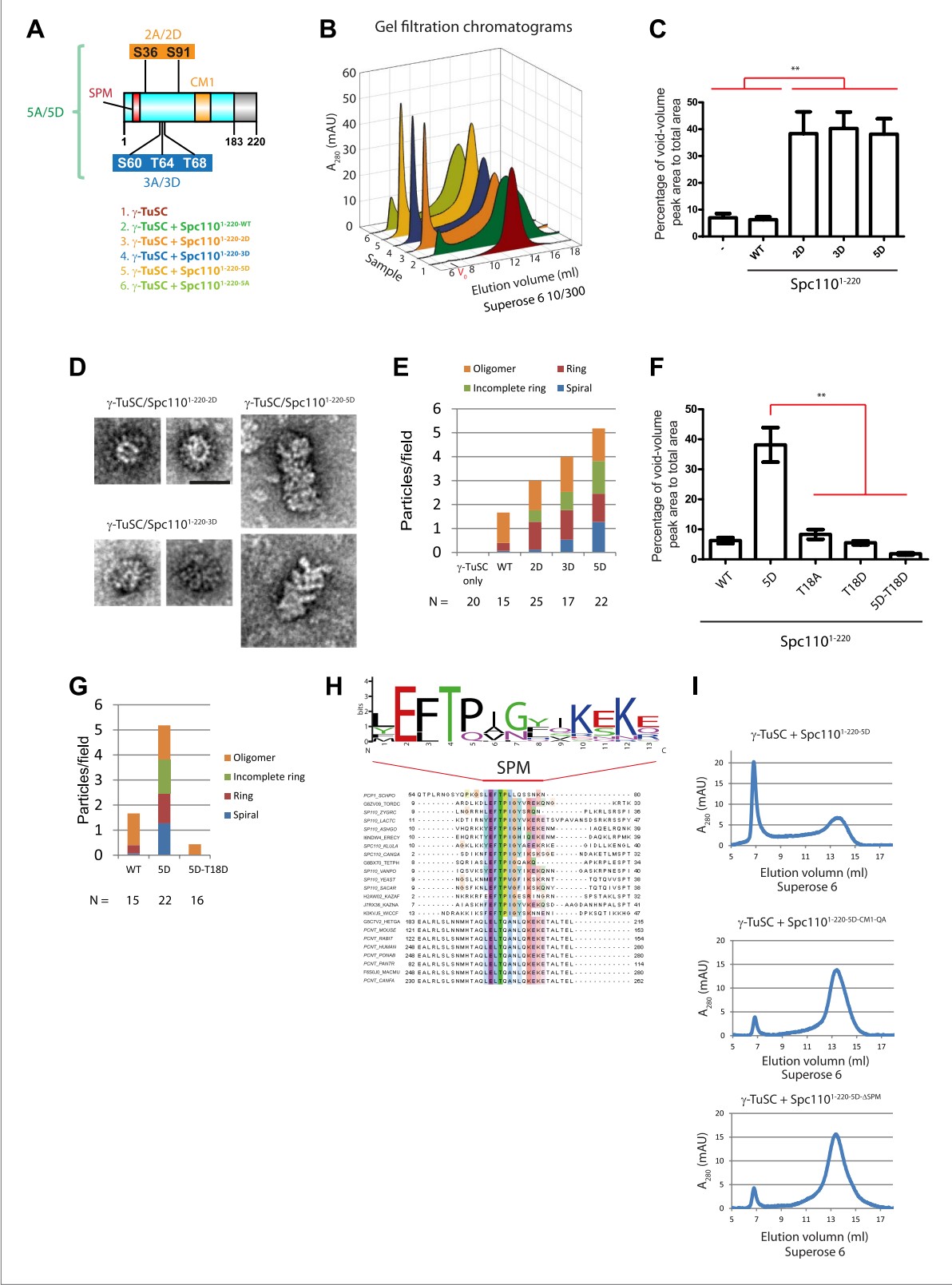

**Figure 2.** Mps1 and Cdk1 phosphorylation of N-Spc110 stimulates γ-TuSC oligomerization. (**A**) Summary of combination of non-phosphorylatable and phospho-mimicking mutations in Spc110 used in this study. The indicated Spc110$^{1-220}$ variants were expressed, purified from *E. coli*, and then tested for γ-TuSC oligomerization. WT: wild-type; 2A/D: S36A/D, S91A/D; 3A/D: S60A/D, T64A/D, T68A/D; 5A/D: S36A/D, S91A/D, S60A/D, T64A/D, T68A/D
*Figure 2. Continued on next page*

*Figure 2. Continued*

(see *Figure 2—figure supplement 1A* for SDS-PAGE of purified Spc110$^{1–220}$ proteins and *Figure 2—figure supplement 1C* for gel filtration chromatograms). (**B**) Spc110$^{1–220}$ phospho-mimicking proteins induced γ-TuSC oligomerization. Spc110$^{1–220}$ proteins were incubated with γ-TuSC in TB150 buffer. The reconstituted complexes were separated according to size by gel filtration using a Superose 6 column. (**C**) Bar graph of area ratio of void-volume peak to total area of chromatogram of (**B**). ** marks statistical significance at p<0.01. Error bars represent SEM. N = 3 to 9 for the number of experiments performed. (**D**) Void-volume peak fractions of (**B**) were subjected to negative staining and protein complexes were analyzed with electron microscopy. Representative ring-like structures of γ-TuSC-Spc110$^{1–220-2D}$, γ-TuSC-Spc110$^{1–220-3D}$, and γ-TuSC-Spc110$^{1–220-5D}$. See *Figure 2—figure supplement 4A* for additional EM pictures. Scale bar: 50 nm. (**E**) Quantification of (**D**). Shown is the particle number per field. Particles were categorized based on the morphology. N is indicated on the figure for the number of fields analysed. (**F**) Quantification of void-volume fractions of γ-TuSC chromatograms. γ-TuSC was incubated with Spc110$^{1–220-WT}$, Spc110$^{1–220-5D}$, Spc110$^{1–220-T18A}$, Spc110$^{1–220-T18D}$, and Spc110$^{1–220-5D–T18D}$ as described in (**B**). Bar graph of area ratio of void-volume peak to total area of chromatogram was calculated as in (**C**). ** marks statistical significance at p<0.01. Error bars represent SEM. N = 3 to 9 for the number of experiments performed. (**G**) Quantification of EM. Void-volume peak fractions of (**F**) were subjected to negative staining and protein complexes were analyzed with electron microscopy. Particles were categorized based on the morphology. Shown is the particle number per field. Note, the Spc110$^{1–220–WT}$ and Spc110$^{1–220-5D}$ graphs are the same as in (**E**). N is indicated on the figure for the number of fields analysed. (**H**) Multiple sequence alignment of SPM element of γ-complex receptors from yeast to human. Residues are marked according to the ClustalX colour scheme. The occurrence of each amino acid in each position of CM1 motif is presented with Weblogo 2.0. (**I**) The indicated Spc110$^{1–220}$ proteins (Spc110$^{1–220-5D}$, Spc110$^{1–220-5D-CM1-QA}$, and Spc110$^{1–220-5D-ΔSPM}$) were incubated with γ-TuSC in TB150 buffer. The reconstituted complexes were separated according to size by gel filtration using a Superose 6 column.

The following figure supplements are available for figure 2:

**Figure supplement 1**. Purification of Spc110$^{1–200}$ variants.

**Figure supplement 2**. Phosphomimetic but not SPM defective N-Spc110 proteins induce oligomerization of γ-TuSC.

**Figure supplement 3**. Non-phosphorylatable mutations in Mps1 or Cdk1 sites of Spc110 are neutral to γ-TuSC oligomerization induced by phosphomimetic mutations.

**Figure supplement 4**. EM single particle analysis of oligomerized γ-TuSC.

**Figure supplement 5**. Mutations of T18 abolish the γ-TuSC oligomerization promoting activity by inactivating the SPM motif.

**Figure supplement 6**. Multiple sequence alignment of CM1 motif-containing proteins.

γ-TuSC ring-like assemblies were observed when Spc110$^{1–220-2D}$, Spc110$^{1–220-3D}$, and Spc110$^{1–220-5D}$ (*Figure 2D,E*, *Figure 2—figure supplement 4A,B*) had been added to the γ-TuSC preparation. Interestingly, the percentage of γ-TuSC spirals increased steadily with increasing numbers of phosphomimetic mutations from Spc110$^{1–220-2D}$, Spc110$^{1–220-3D}$ to Spc110$^{1–220-5D}$ (*Figure 2E*, *Figure 2—figure supplement 4B*), while the number of rings per field was relatively constant (*Figure 2—figure supplement 4C*). The outer diameter of the ring-like assemblies was on average around 40 nm (*Figure 2—figure supplement 4D*). Thus, combined phosphorylation of Spc110 by Cdk1 and Mps1 kinases enhances the oligomerization activity of Spc110 better than phosphorylation by either kinase alone.

## The SPM motif is important for Spc110 oligomerization promoting activity

An additional phosphorylation of N-Spc110 at T18 has been identified in phosphoproteomic studies (*Albuquerque et al., 2008*; *Keck et al., 2011*; *Lin et al., 2011*). However, its precise function remained unclear (*Liang et al., 2013*). To elucidate the role of Spc110$^{T18}$ phosphorylation, Spc110$^{1–220-T18D}$, Spc110$^{1–220-2D-T18D}$ (Cdk1 plus T18), Spc110$^{1–220-3D-T18D}$ (Mps1 plus T18), Spc110$^{1–220-5D-T18D}$ (T18D in addition to Mps1 and Cdk1 sites), and Spc110$^{1–220-5D-T18A}$ proteins were purified (*Figure 2—figure supplement 1A*) and tested for their γ-TuSC oligomerization promoting activity. Spc110$^{1–220}$ species with T18D or T18A failed to induce γ-TuSC oligomerization even when combined with the activating 5D mutations (*Figure 2F*, *Figure 2—figure supplement 2C*). Similar data were obtained with Spc110$^{1–220-5D-T18E}$ and Spc110$^{1–220-5D-T18V}$ (*Figure 2—figure supplement 5A*). Moreover, *spc110$^{T18D}$*, *spc110$^{T18E}$*, *spc110$^{T18A}$*, and *spc110$^{T18V}$* showed identical growth defects at 37°C (*Figure 2—figure supplement 5B*). Since the effects of T18D/E and T18A/V mutations were indistinguishable, we cannot attribute these mutations as phosphomimetic or non-phosphorylatable. However, our results strongly suggest that these mutations

affect the structure of an important yet unappreciated element around T18 that is important for γ-TuSC oligomerization. Consistently, with EM we observed hardly any oligomeric structures when γ-TuSC was incubated with Spc110$^{1–220\text{-}5D\text{-}T18D}$ (*Figure 2G*).

With the sequence alignments of Spc110 orthologs from yeast to human, we observed a conserved motif upstream of CM1, which we designated as Spc110/Pcp1 motif (SPM) (*Figure 2H*). Spc110$^{T18}$ sits within this motif. To evaluate the relative importance of CM1 and SPM, we constructed the Spc110$^{1–220\text{-}CM1\text{-}QA}$ mutant protein with mutations (KE to QA) in two highly conserved residues of the CM1 (*Figure 2—figure supplement 6*) and the Spc110$^{1–220\text{-}ΔSPM}$ lacking the SPM motif (deletion of amino acids 1–20). Spc110$^{1–220}$ proteins were incubated with γ-TuSC in TB150 buffer to examine their oligomerization capacity. Spc110$^{1–220\text{-}5D\text{-}CM1\text{-}QA}$ and Spc110$^{1–220\text{-}5D\text{-}ΔSPM}$ mutant proteins failed to activate γ-TuSC oligomerization even when the five activating Cdk1 and Mps1 mutations were present (*Figure 2I*). Thus, Spc110$^{1–220\text{-}5D\text{-}ΔSPM}$ behaved as Spc110$^{1–220\text{-}5D\text{-}T18D}$ (compare *Figure 2I* with *Figure 2—figure supplement 2C*), further emphasizing that T18D inactivates the SPM. Taken together, SPM and CM1 of Spc110 are both required for γ-TuSC oligomerization.

## The SPM and CM1 motifs and the Cdk1 and Mps1 phosphorylations regulate the affinity of Spc110 to γ-TuSC

Phosphorylation of Spc110 may regulate the γ-TuSC oligomerization by changing its binding affinity. Therefore we performed GST-pulldown assay to measure the binding of the Spc110$^{1–220}$ variants to γ-TuSC. A constant amount of γ-TuSC was incubated with increasing concentrations of Spc110$^{1–220}$ variants (0–300 nM). Consistent with the γ-TuSC oligomerization assay (*Figure 2*), Spc110$^{1–220\text{-}5D}$ showed stronger (p=0.0017 for Spc97/Spc98 and p=0.0006 for Tub4) γ-TuSC binding than Spc110$^{1–220\text{-}WT}$ (*Figure 3A,B*). Interestingly, while Spc110$^{1–220\text{-}2D}$ and Spc110$^{1–220\text{-}3D}$ induced comparable levels of γ-TuSC oligomerization (*Figure 2B*), Spc110$^{1–220\text{-}2D}$ showed less γ-TuSC binding than Spc110$^{1–220\text{-}3D}$. A likely explanation is the 30-fold reduction in concentration that was used in the GST-pulldown assay compared to γ-TuSC oligomerization assays. These results suggested that phosphorylation of Spc110 on S36, S60, T64, T68, and S91 promotes γ-TuSC binding to Spc110.

We also analyzed binding of SPM (T18D and ΔSPM) and CM1 (CM1-QA) deficient Spc110$^{1–220}$ mutant proteins to γ-TuSC. Both Spc110$^{1–220\text{-}5D\text{-}T18D}$ and Spc110$^{1–220\text{-}5D\text{-}CM1\text{-}QA}$ exhibited reduced γ-TuSC binding relative to Spc110$^{1–220\text{-}5D}$ (p<0.01 for Spc97/Spc98 and p<0.05 for Tub4) (*Figure 3C,D*). The γ-TuSC binding capacity of Spc110$^{1–220\text{-}5D\text{-}T18D\text{-}CM1\text{-}QA}$ was further reduced (p<0.01, *Figure 3C,D*), suggesting both CM1 and SPM are important for γ-TuSC binding. Interestingly, while Spc110$^{1–220\text{-}5D\text{-}T18D}$, Spc110$^{1–220\text{-}5D\text{-}CM1\text{-}QA}$, and Spc110$^{1–220\text{-}ΔSPM}$ all failed to oligomerize γ-TuSC (*Figure 2F,I*), they showed different levels of γ-TuSC binding capacity (*Figure 3D*). The decoupling of γ-TuSC binding capacity from γ-TuSC oligomerization suggests that both processes are not inevitably linked. Taken together, we conclude that both CM1 and SPM motifs contribute to γ-TuSC binding and are required for promoting γ-TuSC oligomerization.

## SPM and the phosphorylation of N-Spc110 control MT nucleation activity in vitro

In vitro MT nucleation assays were performed to test the effect of Spc110 variants on γ-TuSC-mediated MT nucleation. Since phosphorylation of Spc110 and the presence of a functional SPM regulate γ-TuSC oligomerization into template rings, we expected to see differences in MT nucleation depending on these parameters (*Figures 2 and 3*). Compared to the buffer control and γ-TuSC alone, a ~threefold increase (p<0.05) in MT nucleation activity was observed for Spc110$^{1–220\text{-}2D}$, Spc110$^{1–220\text{-}3D}$, or Spc110$^{1–220\text{-}5D}$ upon incubation with γ-TuSC (*Figure 4A,B*). In contrast, Spc110$^{1–220\text{-}WT}$ showed levels of MT nucleation that were comparable to the buffer control and γ-TuSC alone. Consistent with the proposed role of SPM, the MT nucleation level was reduced to buffer control levels when Spc110$^{1–220\text{-}5D\text{-}T18D}$ was incubated with γ-TuSC (p=0.99). These results are in accordance with EM particle quantification assays, as Spc110$^{1–220\text{-}2D}$, Spc110$^{1–220\text{-}3D}$, or Spc110$^{1–220\text{-}5D}$ induced similar and significantly more γ-TuSC ring assemblies than Spc110$^{1–220\text{-}WT}$ and Spc110$^{1–220\text{-}5D\text{-}T18D}$ (p<0.05, *Figure 2E*, *Figure 2—figure supplement 4C*). In summary, these data support the model that Cdk1 and Mps1 phosphorylations of Spc110 regulate γ-TuSC-mediated MT nucleation by controlling template assembly.

## Phosphorylation of Spc110 is cell cycle dependent

To analyze cell cycle dependent phosphorylation of Spc110, we generated one phospho-specific antibody against the two Cdk1 sites (Spc110$^{S36}$ and Spc110$^{S91}$) (*Figure 5A*) and another against the Mps1

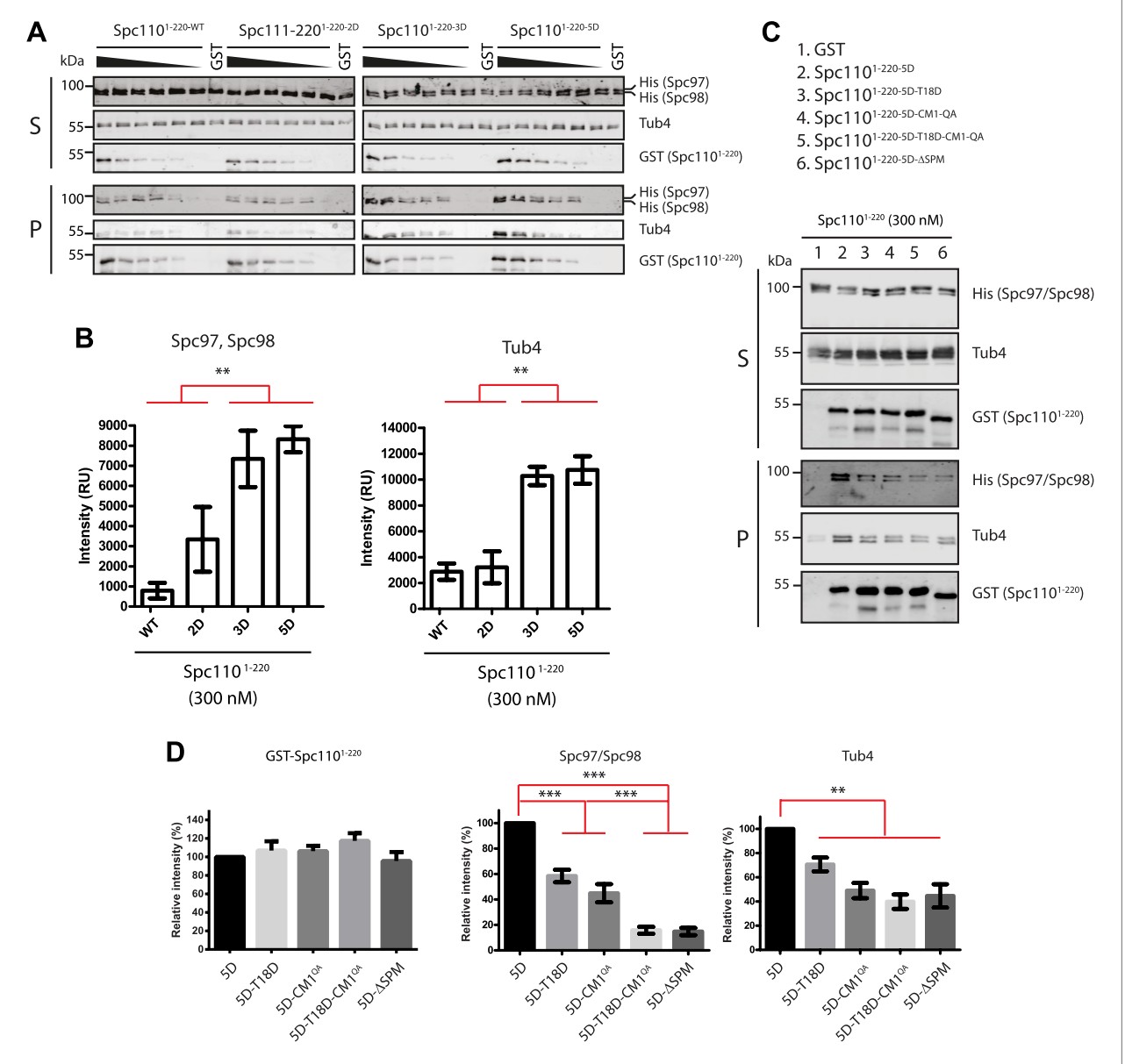

**Figure 3**. Phosphorylation of N-Spc110 regulates the affinity to γ-TuSC. (**A**) GST pull-down assays were performed between γ-TuSC (containing His-tagged Spc97-6His and Spc98-6His) and the GST-tagged Spc110$^{1-220}$ proteins. The bound proteins were eluted with sample buffer and separated on SDS-PAGE and analyzed by immunoblotting with anti-His, anti-GST, and anti-Tub4 antibodies. Infrared-dye-labelled secondary antibodies were applied and detected with Li-cor imaging system. (**B**) Quantification for bound Spc97, Spc98, and Tub4 from (**A**) at 300 nM Spc110$^{1-220}$. ** marks statistical significance at p<0.01. Error bars represent SEM. N = 3 for the number of experiments performed. (**C**) Mutations in SPM or CM1 affect γ-TuSC binding. As (**A**) but performed with the indicated SPM and CM1 mutant Spc110$^{1-220}$ constructs. (**D**) Quantification of Spc110$^{1-220}$ variants and bound Spc97, Spc98, and Tub4 from (**C**) at 300 nM Spc110$^{1-220}$. ** marks statistical significance at p<0.01 and *** at p<0.001. Error bars represent SEM. N = 3 for the number of experiments performed.

sites on Spc110 (Spc110$^{pS60-pT64-pT68}$) (**Figure 5B**). Both anti-Spc110$^{pS36-pS91}$ and anti-Spc110$^{pS60-pT64-pT68}$ antibodies gave specific signals in vitro and in vivo (**Figure 5A–C**).

To understand by which Cdk1 kinase complex Spc110$^{S36}$ and Spc110$^{S91}$ were phosphorylated, we co-immunoprecipitated Cdk1$^{as1}$ either in complex with Clb5-TAP (S phase cyclin) or Clb2-TAP (mitotic cyclin) from yeast lysates and performed in vitro kinase assays using Spc110$^{1-220-WT}$ that had been purified from *E. coli*. Spc110$^{S36-S91}$ was phosphorylated equally by Cdk1-Clb2 and Cdk1-Clb5 as

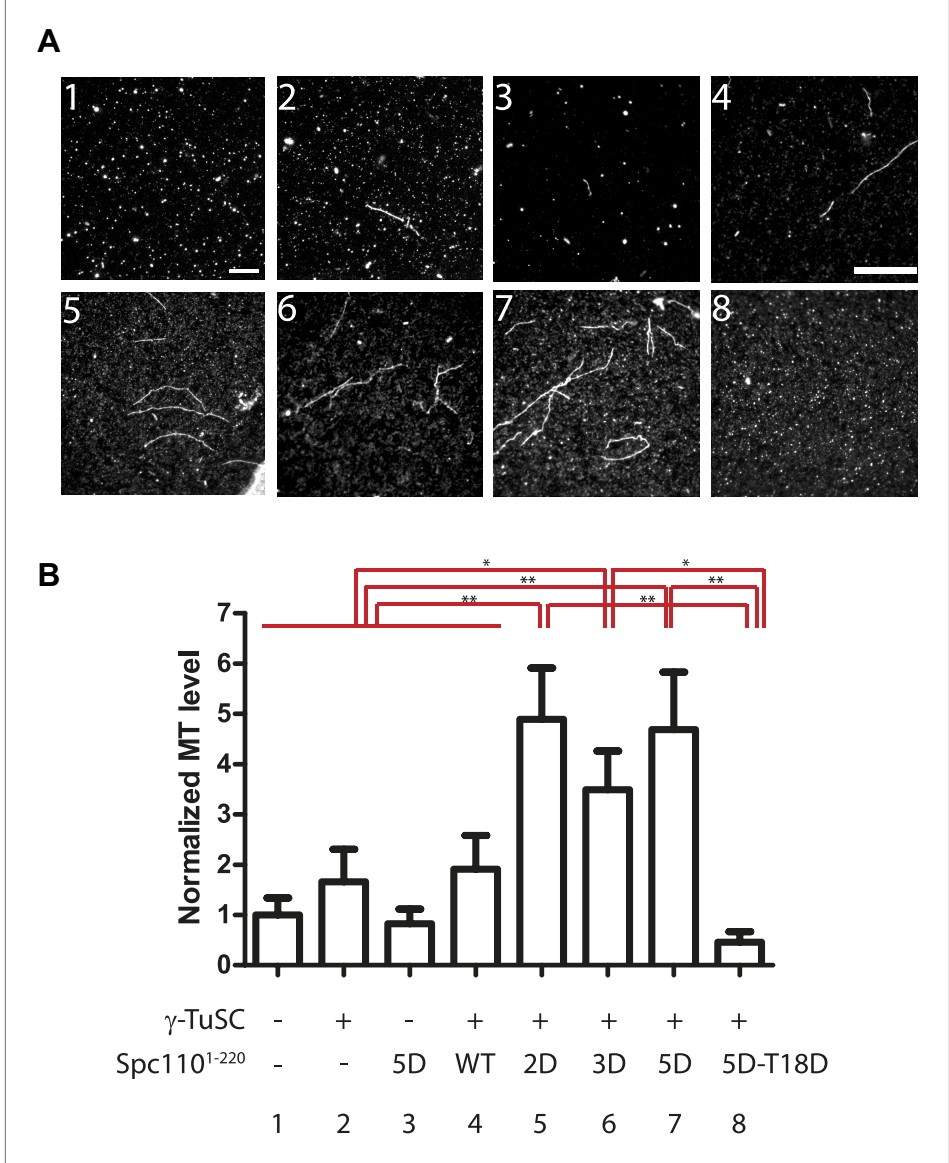

**Figure 4**. Spc110$^{1-220}$ phosphorylation enhances MT nucleation activity in vitro. (**A**) Enhancement of MT nucleation activity of Spc110 by Mps1 and Cdk1 phosphorylation. Representative image fields of Alexa546-labeled microtubules from the nucleation assay polymerized in the presence of buffer, γ-TuSC and Spc110$^{1-220}$ variants (**B**). Scale bar: 10 μm. (**B**) Quantification of the MT nucleation assay. For each experiment, the number of MTs was counted from 20 fields. The MTs/field was normalized with MTs/field obtained by the MT nucleation reaction in the presence of buffer only. N = 6 to 9 for the number of independent experiments performed. * marks statistical significance at $p < 0.05$ and ** at $p < 0.01$. Error bars represent SEM.

detected by Spc110$^{pS36-pS91}$ antibodies (*Figure 5A*). It is important to note that both Cdk1 kinase complexes had equal kinase activities toward histone H3 (*Figure 5A*, bottom).

To confirm the cell-cycle dependent phosphorylation of Spc110 in vivo, we immuno-precipitated (IP) Spc110-GFP with GFP-binder from cells arrested either in G1 phase with α-factor, S phase with hydroxyurea (HU) or prometaphase with nocodazole, and then blotted with phospho-specific antibodies. At G1 phase, no signal was observed with any of the two phospho-specific antibodies (*Figure 5D*). The phosphorylation of Mps1 was detected at S phase and maintained in mitosis. The Spc110$^{pS36-pS91}$ signal also appeared at S phase, and continuously increased in mitosis.

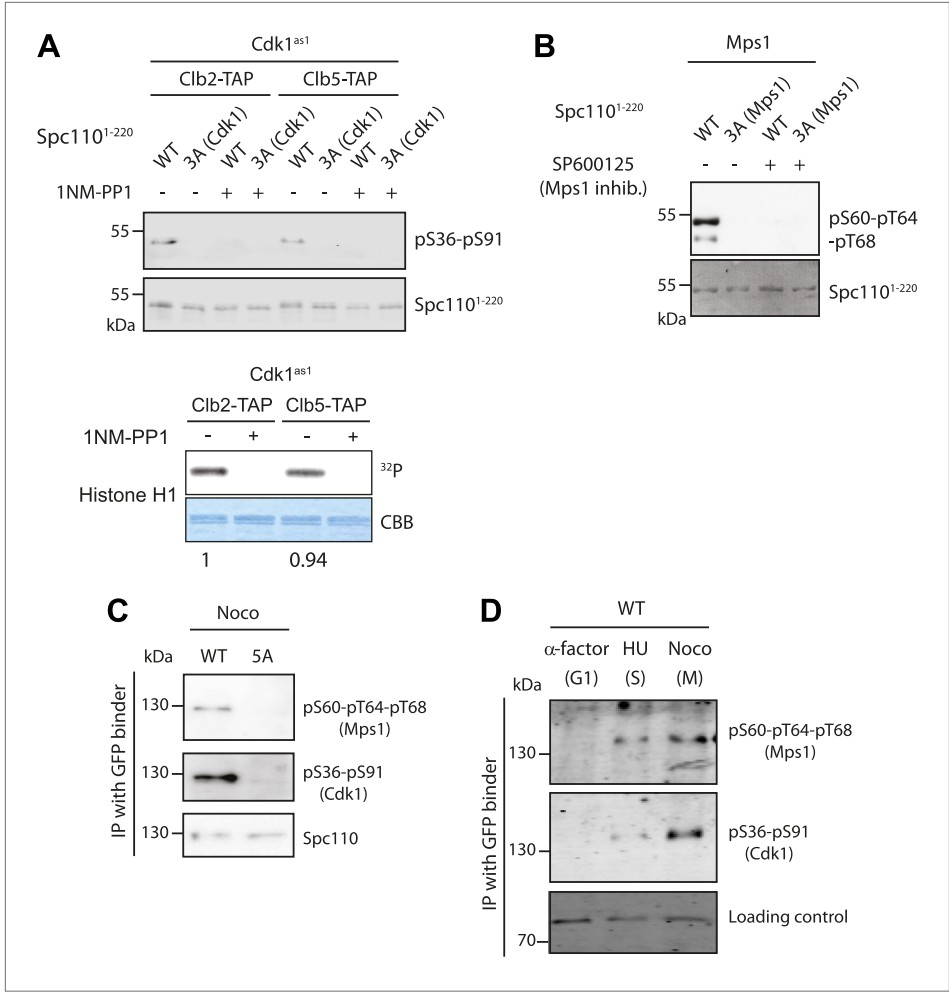

**Figure 5**. Cell cycle dependent phosphorylation of Spc110. (**A** and **B**) Two phospho-specific antibodies were generated from guinea pigs to recognize phosphorylation of Cdk1 sites (pS36-pS91) (**A**) and Mps1 sites (p60-p64-pT68) (**B**). In vitro kinase assays were performed in the presence of recombinant Mps1 or Cdk1as1 and Spc110$^{1-220}$, either wild type (WT) or non-phosphorylatable variants as indicated. 3A (Cdk1) indicates T18A-S36A-S91A and 3A (Mps1) S60A-T64A-T68A. In vitro phosphorylated Spc110$^{1-220}$ was subjected to SDS-PAGE and immunoblot with the corresponding phospho-specific antibodies. The specific kinase activity of Cdk1as1-Clb2 and Cdk1as1-Clb5 was compared using human histone H1 as substrate (**A**, bottom). The numbers in the histone H1 experiment represent the relative kinase activity. As negative control of kinase activity, Mps1 was inactivated with chemical inhibitor SP600125 (**B**), while Cdk1as1-Clb2 and Cdk1as1-Clb5 overexpressed and purified from budding yeast were inactivated with 1NM-PP1 (**A**). (**C**) Phosphorylation of Spc110 and Spc110$^{5A}$ in vivo. *SPC110-GFP* wild type cells (WT) and *spc110$^{5A}$-GFP* cells were arrested in mitosis with nocodazole. Spc110-GFP was enriched using GFP binder conjugated to beads, and the bound proteins were subject to immunoblotting with the indicated antibodies. (**D**) *SPC110-GFP* wild type cells were arrested in G1, S phase, and mitosis as indicated. Spc110-GFP was enriched with GFP-binder and analyzed by immunoblotting with the indicated P-specific antibodies. A non-specific band was used as loading control.

The following figure supplements are available for figure 5:

**Figure supplement 1**. Phosphorylation of T18 most likely affects γ-TuSC oligomerization promoting activity of Spc110 in a negative manner.

T18 within the SPM motif has been found phosphorylated in vivo and matches the minimal Cdk1 consensus sequence S/T-P (*Keck et al., 2011*). To elucidate the function of T18 phosphorylation, we raised phospho-specific antibodies against pT18. In vitro phosphorylation assay showed that T18 was only phosphorylated by Cdk1-Clb2 but not by Cdk1-Clb5 (*Figure 5—figure supplement 1A*). Due to

the low sensitivity of anti-Spc110$^{pT18}$, we used mass spectrometry for the analysis of T18 phosphorylation in vivo. Total yeast cell lysates were fractionated into supernatant and pellet that contained SPBs. The pellet fraction was solubilized with high salt buffer and then used for Spc110 enrichment. pT18 was predominately detected in the SPB-containing pellet fraction of prometaphase cells but not in the corresponding fraction of G1 or S phase arrested cells (*Figure 5—figure supplement 1B,C*). These data suggest that T18 of Spc110 becomes phosphorylated at SPBs in M phase but not S phase, which is consistent with published data (*Keck et al., 2011*).

Finally, we addressed the effects of T18 phosphorylation on γ-TuSC oligomerization. Because the T18A/V/D/E mutations inactivated the SPM function, we turned to an in vitro phosphorylation approach. Spc110$^{1–220-5D}$ was incubated with Cdk1$^{as1}$-Clb2 in the presence or absence of the Cdk1$^{as1}$ inhibitor 1NM-PP1. Only in the absence of 1NM-PP1, Spc110$^{1–220-5D}$ was phosphorylated on T18 as shown by the phospho-specific anti-T18 antibodies. We incubated Spc110$^{1–220-5D}$ and Spc110$^{1–220-5D-pT18}$ with γ-TuSC and analyzed γ-TuSC by gel filtration. The activity of Spc110$^{1–220-5D-pT18}$ to promote γ-TuSC oligomerization was significantly reduced in comparison with Spc110$^{1–220-5D}$ ($p<0.05$) (*Figure 5—figure supplement 1D,E*). This suggests that in vitro phosphorylation of T18 by Cdk1-Clb2 inhibits γ-TuSC oligomerization.

Taken together, our results are in agreement with previous reports about the timing of Spc110 phospho-dependent mobility shift in SDS-PAGE (*Friedman et al., 1996*; *Stirling and Stark, 1996*) and suggest that the Cdk1 (S36, S91) and Mps1 sites become phosphorylated in S and in mitosis. In addition, phosphorylation of T18 within SPM occurs in mitosis and may negatively regulate SPM function.

## Cells with *spc110* phospho-, SPM-, and CM1-mutant alleles have SPBs with impaired MT nucleation activity

To reveal the function of phosphorylation of Spc110 in vivo, we replaced *SPC110* with phosphomimetic and non-phosphorylatable mutant alleles. There was no significant difference in the expression levels of the Spc110 variants (*Figure 6—figure supplement 1A*). *spc110$^{2A/2D}$*, *spc110$^{3A/3D}$*, *spc110$^{5A/5D}$*, the SPM deficient *spc110$^{T18D}$*, *spc110$^{ΔSPM}$*, and the CM1 deficient *spc110$^{CM1-QA}$* cells grew at 23°C comparable to wild-type *SPC110* (WT) cells. However, *spc110$^{T18D}$* and *spc110$^{ΔSPM}$* cells displayed a conditional lethal growth defect at 37°C and the SPM/CM1 defective *spc110$^{T18D-CM1-QA}$* double mutant was non-viable at 23°C and 37°C (*Figure 6A*, left panel). Combining T18D with 2D (Cdk1 sites), 3D (Mps1 sites) and 5D mutations slightly reduced growth at 37°C compared to the 2D, 3D, and 5D mutations (*Figure 6A*). This shows that, consistent with our in vitro γ-TuSC oligomerization results (*Figure 2—figure supplement 2C*), the SPM mutation T18D is dominant over the activating Cdk1 and Mps1 phosphomimetic mutations.

The defect of some mutant alleles involved in spindle assembly becomes apparent only after SAC function has been impaired (*Wang and Burke, 1995*; *Daum et al., 2000*). In the absence of the SAC gene *MAD2* (*Li and Murray, 1991*), slow growth was observed in *spc110$^{2A}$* and *spc110$^{2D}$* cells at 37°C. This growth defect was even more pronounced in *spc110$^{5A}$ mad2Δ* cells at 37°C (*Figure 6A*, right panel). All *spc110 mutant* alleles with SPM or CM1 inactivating mutations showed reduced growth in the *mad2Δ* background compared to *MAD2* SAC proficient cells (*Figure 6A*, bottom panel). Together, these growth tests suggest that the absence of phospho-regulation of Spc110 or the lack of SPM or CM1 results in defects that are compensated by a SAC induced delay in cell cycle progression.

Based on our model built from in vitro experiments, phosphorylation of N-Spc110 by Cdk1 (S36, S91) and Mps1 in S phase promotes the binding and oligomerization of γ-TuSC to form the template required for MT nucleation. In addition, the SPM and the CM1 motifs of Spc110 are important for γ-TuSC oligomerization and MT nucleation (*Figure 2*). Thus, mutations in *SPC110* that affect any one of these sites/motifs should display a reduction in their SPB-associated tubulin signal. To test this hypothesis, we expressed *GFP-TUB1* in cells and measured the intensity of the tubulin signal at the SPB at different cell cycle phases. Cells were synchronized with α-factor in G1 and then released into prewarmed media at 37°C, the restrictive temperature of some of the mutant cells. In SPM defective *spc110$^{T18A}$* and *spc110$^{T18D}$* cells, the reduction of GFP-tubulin signal at SPBs was already apparent in G1 (*Figure 6B*, *Figure 6—figure supplement 1B*, $p<0.001$). In S phase, the stage where newly synthesized SPBs mature and MT nucleation is initiated, all *spc110* mutants except for *spc110$^{5D}$* cells exhibited a marked reduction in MT nucleation at SPBs when compared with WT cells (*Figure 6B*; for *spc110$^{5A}$*, $p<0.05$; for *spc110$^{T18D}$*, *spc110$^{T18A}$*, $p<0.001$; *Figure 6—figure supplement 1B*). The reduction in tubulin signal was especially pronounced in *spc110$^{T18D}$* and *spc110$^{T18A}$* cells, indicating that SPM mutations blocked MT formation in vivo. *spc110$^{ΔSPM}$* cells also showed MT defects at 37°C (*Figure 6—figure supplement 1F*).

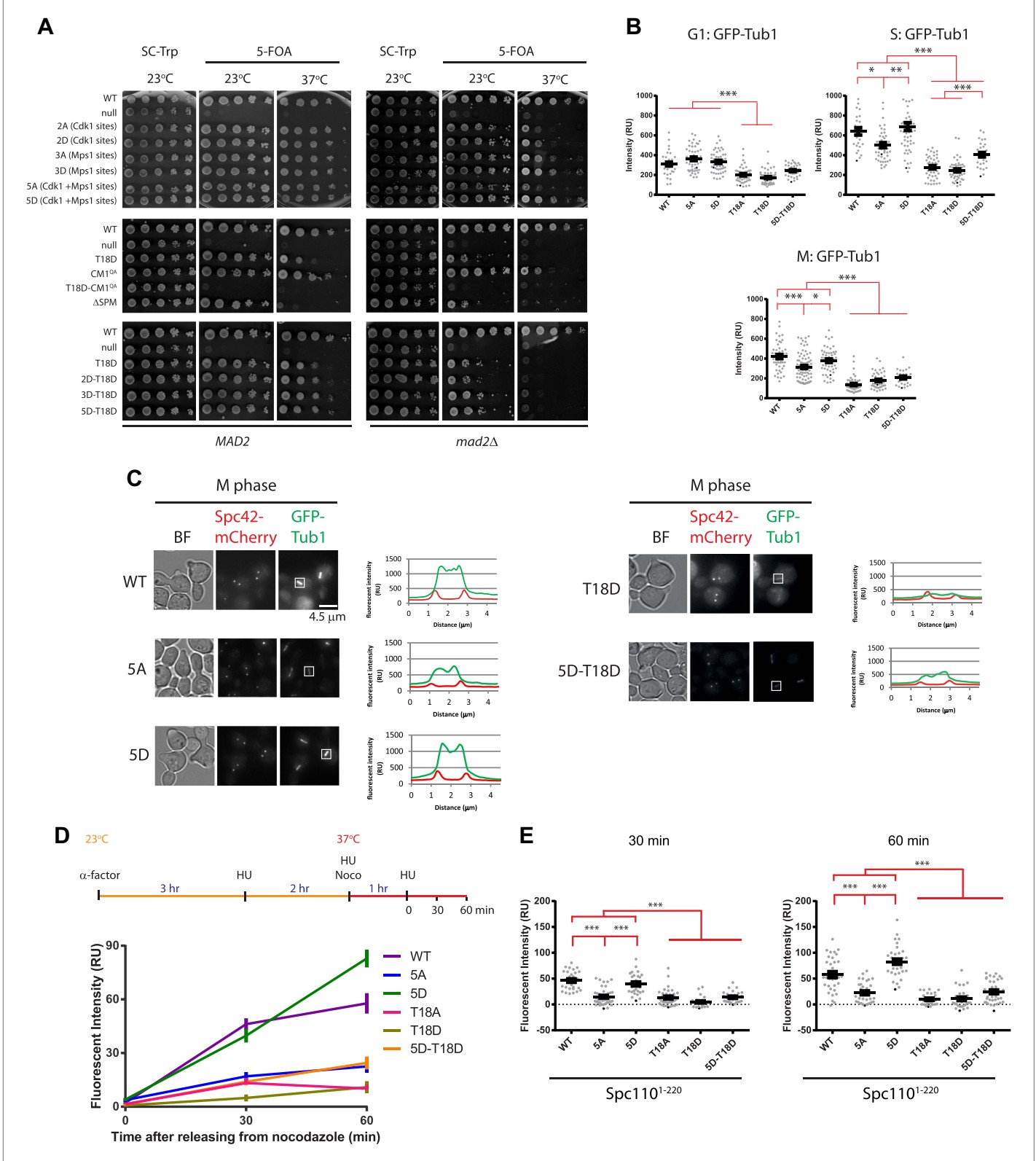

**Figure 6**. Cells with *spc110* phospho-, SPM-, or CM1-mutant alleles have defects in spindle formation. (**A**) Growth of 10-fold serial dilutions of *SPC110* shuffle strains with integration vector encoding *SPC110* (WT) or *SPC110* mutants with and without the SAC gene *MAD2*. Growth was tested either on synthetic complete (SC) plates containing 5-FOA or SC dropout plates. (**B**) The indicated *SPC110* wild type cells and *spc110* mutant cells carrying

*Figure 6. Continued on next page*

*Figure 6. Continued*

*GFP-TUB1 SPC42-mCherry* were incubated at 37°C, the restrictive temperature of some of the *spc110* mutants (***Figure 6A***). The fluorescent signal of GFP-Tub1 at SPBs was quantified in G1 phase cells (no buds), early S phase cells (small bud with unsplit SPBs) and M phase cells (large bud with split SPBs and spindle length within 2 μm). 50 cells were analyzed per cell cycle phase and strain. Error bars represent SEM. * marks statistical significance at p<0.05, ** at p<0.01, and *** at p<0.001. N = 2 or 3 for the number of experiments performed. (**C**) The representative metaphase cells from (**B**) were scanned for the distribution of the GFP-Tub1 and the Spc42-mCherry signal along the spindle axis. Scale bar: 4.5 μm. (**D**) In vivo MT re-nucleation assay. Wild-type *SPC110* cells (WT) and the indicated *spc110* cells with *GFP-TUB1* were treated as indicated in the outline. The GFP-Tub1 signal at SPBs was measured at 0, 30, and 60 min after nocodazole washout. N = 40 for the number of cells analyzed per time point and strain. Error bars represent SEM. (**E**) Data of (**D**) after 30 and 60 min. Error bars represent SEM. *** marks statistical significance at p<0.001.
The following figure supplements are available for figure 6:

**Figure supplement 1**. Phenotype of *SPC110* mutants.

The situation in metaphase was similar to that of S phase although the overall relative differences between WT and *spc110* mutants were less pronounced than in S phase (***Figure 6B***, p<0.001). Consistently, the overall spindle MT signal was reduced in *spc110*[5A], *spc110*[T18A], *spc110*[T18D], and *spc110*[5D-T18D] cells (***Figure 6B,C***). Taken together, the in vivo analysis of the tubulin signal at SPBs in *spc110* cells is consistent with the in vitro data on γ-TuSC oligomerization and MT nucleation (***Figures 2 and 4***).

Reduced MT signal intensity at SPBs would be indicative of impaired γ-TuSC activity. In the simplest interpretation, γ-TuSC recruitment to SPBs is impaired. We found that the SPB signal from the γ-TuSC marker Spc97-GFP was reduced in *spc110*[5A], *spc110*[T18D], *spc110*[5D-T18D], and *spc110*[ΔSPM] cells in G1 and S phase compared to *SPC110*[WT] and *spc110*[5D] cells (***Figure 6—figure supplement 1C,G***). However, this reduction was not observed in mitotic cells. Analysis of Spc97-GFP signal intensity at SPBs in *SPC110* wild type, *spc110*[T18D], and *spc110*[ΔSPM] cells revealed that the SPM mutation allowed Spc97 SPB binding but with a delay in cell cycle timing (***Figure 6—figure supplement 1D,G***). After SPB separation in late S phase wild type, *spc110*[T18D] and *spc110*[ΔSPM] cells eventually carried similar amounts of Spc97-GFP at SPBs.

Analysis of the GFP signal at SPBs in *SPC110-GFP*, *spc110*[5A]-GFP *spc110*[T18D]–GFP, and *spc110*[5D-T18D]-GFP cells did not reveal any differences between cell types (***Figure 6—figure supplement 1E***). Thus, the reduction of the Spc97-GFP signal in G1 and S phase most likely reflects a reduction of γ-TuSC binding to the *SPC110* mutant molecules at SPBs. The observation that all *spc110* cells had similar γ-TuSC signals at mitotic SPBs while the MT signal was only reduced in *spc110*[T18D] and *spc110*[5A] cells (***Figure 6—figure supplement 1C*** vs ***Figure 6B***), suggests that the SPB-associated γ-TuSC is not fully active in MT organization in these cells. The functions of Spc110 probably extend beyond γ-TuSC binding, most likely to include γ-TuSC oligomerization ('Discussion'). However, compared with *spc110*[T18D] or *SPC110* cells, Spc110-GFP at the SPB was reduced in *spc110*[ΔSPM] cells (***Figure 6—figure supplement 1H***), suggesting the *SPM* deletion affects incorporation of Spc110 into the SPB. This may contribute to the *spc110*[ΔSPM] phenotype and may explain why the growth defect of *spc110*[ΔSPM] cells was more pronounced than that of *spc110*[T18D] cells (***Figure 6A***).

To further examine the MT nucleation defect in *spc110* phospho-mutant cells, we performed an in vivo MT nucleation assay. Cells were first arrested in S phase with HU, then MTs were depolymerized with nocodazole, followed by nocodazole wash out to trigger MT nucleation by the SPB bound γ-TuSC (***Erlemann et al., 2012***). To prevent cell cycle progression into anaphase we retained the cells in the HU block (***Figure 6D***). The re-growth of MTs was measured as a change in GFP-Tub1 intensity at SPBs. In comparison to WT and *spc110*[5D] cells, MT nucleation activity was reduced in the *spc110*[5A], *spc110*[T18A], *spc110*[T18D] *spc110*[5D-T18D] cells (***Figure 6E***). Again, these data are consistent with the in vitro data (***Figures 1–4***) and suggest that Spc110's ability to stimulate MT nucleation is regulated through stimulatory phosphorylation and requires the SPM motif.

## Spc110 interaction and γ-TuSC oligomerization are facilitated by the N-terminal region of Spc98

The N-terminal region of Spc98/GCP3 orthologs is conserved. Spc98/GCP3 family members contain five predicted helices followed by an unstructured linker region (***Figure 7A***, ***Figure 7—figure supplement 1A***). The conserved GRIP1 and 2 motifs that are common to all GCP proteins follow this

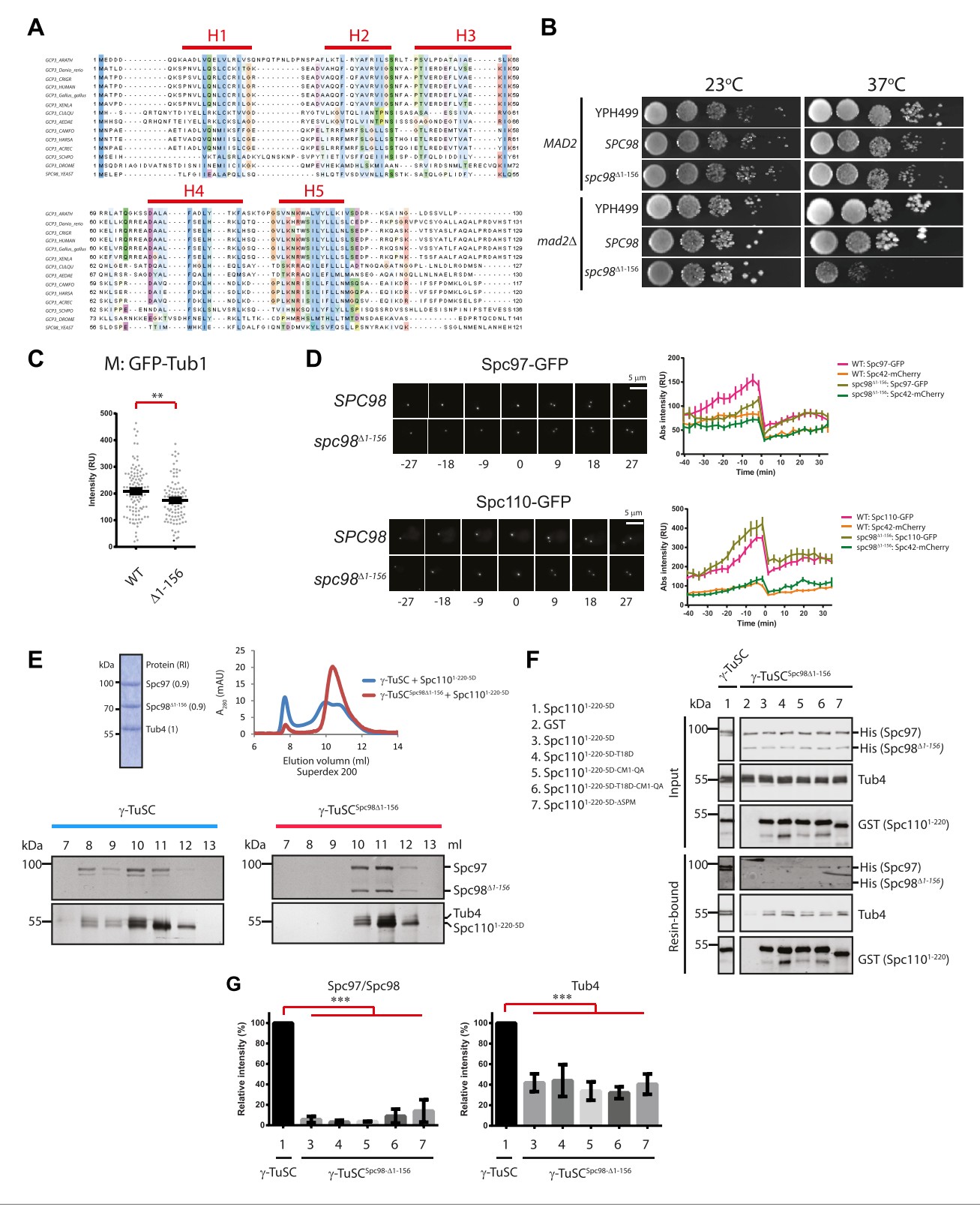

**Figure 7**. The N-terminus of Spc98 mediates binding to N-Spc110. (**A**) Alignment of the amino acid sequence of GCP3 homologues from yeast to human. Shown are the putative α-helical regions H1–H5. Residues are marked according to the ClustalX colour scheme. (**B**) Growth test of *spc98Δ1–156* cells with and without the SAC gene *MAD2* at 23°C and 37°C. "*SPC98*" indicates the *spc98ΔSPC98* cells while YPH499 is the unmodified wild type strain. (**C**) GFP-Tub1 signal

*Figure 7. Continued on next page*

*Figure 7. Continued*

at SPBs in *SPC98* wild type or *spc98[Δ1–156]* cells. The experiment was performed as in *Figure 6B*. ** marks statistical significance at p<0.01. N = 100 cells analysed for each strain. (**D**) *SPC98* and *spc98[Δ1–156]* cells with *SPC42-mCherry*, *SPC97-GFP*, or *SPC110-GFP* were analyzed by time lapse analysis. The fluorescent intensity was measured over time (right). t=0 for the time point of SPB separation. Error bars represent SEM. N = 14 and 19 for the number of cells analysed in *SPC97-GFP* and *SPC-110-GFP* strains respectively. The Spc97-GFP and Spc110-GFP signals of selected cells are shown on the left panel. Scale bar: 5 μm. (**E**) Spc110[5D] combined with γ-TuSC[Spc98Δ1–156] fails to induce oligomerization in TB150 buffer. The experiment was performed as in *Figure 2B*. The purified γ-TuSC[Spc98Δ1–156] after SDS-PAGE and Coomassie Blue staining is shown with relative intensity of the protein bands. The protein distribution in the Superdex 200 10/300 chromatogram was analyzed with SDS-PAGE and silver staining. (**F**) Binding of γ-TuSC and γ-TuSC[Spc98Δ1–156] to Spc110[1–220] mutant proteins. The binding reaction was performed and analyzed by immunoblotting as described in *Figure 3A*. (**G**) Quantification of *Figure 7F*. Pull-downed Spc97, Spc98, and Tub4 proteins at 300 nM Spc110[1–220-5D]. Error bars represent SEM. N = 3 for the number of experiments performed. *** marks statistical significance at p<0.001.

The following figure supplements are available for figure 7:

**Figure supplement 1**. The importance of the N-terminal region of Spc98.

**Figure supplement 2**. γ-TuSC[Spc98Δ1–156] maintains self-oligomerization capability in BRB80 buffer.

N-terminal region (*Wiese and Zheng, 2006*). Analysis of γ-TuSC by electron microscopy has shown that the N-terminus of Spc98 is in close proximity to the N-Spc110 binding site (*Kollman et al., 2010*). This topological arrangement raises the possibility that N-Spc98 is involved in binding to N-Spc110 and oligomerization of γ-TuSC. To test this idea, we constructed truncations of N-Spc98 with which to assess the function and biochemical behaviour of γ-TuSC[Spc98ΔN]. All N-terminal truncations of *SPC98* supported cell growth at 23°C (*Figure 7B*, *Figure 7—figure supplement 1B*). However, Δ1–156, Δ1–177, and ΔLinker deletions of *SPC98* showed reduced or no growth in *mad2Δ* cells at 37°C.

We analyzed the phenotype of *spc98ΔN* mutants at their restrictive temperature, 37°C. Because γ-TuSC[Spc98Δ1–177] formed aggregates already in TB150 buffer and was therefore unsuitable for oligomerization assay (*Figure 7—figure supplement 2A,B*), we focus our analysis on *spc98[Δ1–156]*. Like the wild-type γ-TuSC, the purified recombinant γ-TuSC[Spc98Δ1–156] was monomeric in TB150 (*Figure 7—figure supplement 2C,D*). *spc98[Δ1–156]* cells incubated at 37°C had weaker GFP-Tub1 signal at their SPBs than wild type *SPC98* cells (WT, *Figure 7C*). This is consistent with the observed mild MT nucleation defect of *spc98[Δ1–156]* cells at 37°C (*Figure 7—figure supplement 1C,D*). Moreover, time-lapse analysis of α-factor synchronized *SPC98 SPC42-Cherry* and *spc98[Δ1–156] SPC42-mCherry* cells with *SPC97-GFP* or *SPC110-GFP* revealed a delay in Spc97-GFP recruitment to SPBs at the time of SPB duplication (*Figure 7D*). Interestingly, slightly more Spc110-GFP was found at SPBs in *spc98[Δ1–156]* cells, which may help to compensate the *spc98[Δ1–156]* phenotype. Thus, the N-terminal region of Spc98 is important for optimal MT organization and timely γ-TuSC recruitment to SPBs.

We next asked whether γ-TuSC[Spc98Δ1–156] can be oligomerized by incubation with Spc110[1–220-5D]. Interestingly, Spc110[1–220-5D] failed to shift γ-TuSC[Spc98Δ1–156] to the void volume of the Superose 6 column, as was the case for γ-TuSC (*Figure 7E*). In agreement with the oligomerization experiment, γ-TuSC[Spc98Δ1–156] showed reduced binding to Spc110[1–220-5D] in the pull down assay (*Figure 7F,G*). Thus, the N-terminus of Spc98 is involved in binding to Spc110. In spite of not changing our conclusion, we noticed that Spc98[Δ1–156]-6His was detected less strongly than Spc98-6His by the anti-His antibodies, although in Coomassie Blue stained gels both proteins were present in 1:1 ratio (*Figure 7E*, upper panel and F). Thus, it is likely that Spc98[Δ1–156]-6His transferred less efficiently to the blotting membrane than Spc97-6His, leading to an underestimation of the Spc98[Δ1–156]-6His signal relative to full-length Spc98-6His.

Finally, we tested γ-TuSC[Spc98Δ1–156] oligomerization in the BRB80 buffer that supports oligomerization without the aid of Spc110. γ-TuSC and γ-TuSC[Spc98Δ1–156] shifted to the void volume of the Superose 6 column with equal efficiency (*Figure 1—figure supplement 2B*, *Figure 7—figure supplement 2C*). Analysis of the void fraction by EM identified oligomeric γ-TuSC[Spc98Δ1–156] rings (*Figure 7—figure supplement 2E*). Therefore, the N-terminus of Spc98 is important for Spc110-mediated oligomerization but not for Spc110-independent γ-TuSC oligomerization.

## Bioinformatic analysis of the C-terminal SPB/centrosomal binding domain of SPM-CM1 and CM1-only γ-TuCRs

Our biochemical and cell biology analyses have identified the novel motif SPM and CM1 (*Sawin et al., 2004*) as important motifs in Spc110 for γ-TuSC binding and oligomerization. Using bioinformatics

approaches, we investigated the presence of SPM and CM1 motifs in γ-TuCRs from various species (**Figure 8A,B**). A yet unappreciated CM1 motif was identified in another budding yeast γ-TuCR, Spc72 (**Knop and Schiebel, 1998**; **Figure 8A**, **Figure 8—figure supplement 1C**).

Based on the presence of SPM and CM1 motifs, γ-TuCRs can be divided into the SPM-CM1 and the CM1-only types. Spc110 and fission yeast Pcp1 fall into the SPM-CM1 class. Our analysis also predicts SPM and CM1 motifs, albeit with moderate sequence deviation, in mammalian PCNT (**Figure 8B**). Another group of γ-TuCRs have only a CM1 motif but no SPM. Human CDK5RAP2 and myomegalin, *Drosophila* Cnn, budding yeast Spc72, and fission yeast Mto1 fall into this class (**Figure 8B**).

A further level of category resolution was achieved by the classification based on the C-terminal MTOC targeting domains (**Figure 8—figure supplement 1A,B**). It was shown that yeast Spc110 (**Sundberg et al., 1996**; **Stirling and Stark, 2000**), Pcp1 (**Fong et al., 2010**), *Drosophila* D-PLP (Cp309) (**Kawaguchi and Zheng, 2004**), and mammalian PCNT (**Gillingham and Munro, 2000**; **Takahashi et al., 2002**) are targeted to nuclear side of SPBs or centrosomal MTOCs via the conserved PACT domain (**Figure 9A**). In case of SPBs that are embedded in the nuclear envelope and therefore have a cytoplasmic and a nuclear side, SPM-CM1 γ-TuCRs only function in organizing the nuclear MTs that separate the sister chromatids in mitosis (**Figure 9B**). In contrast, CM1-only γ-TuCRs of fungi, such as Spc72 in budding yeast (**Knop and Schiebel, 1998**), Mto1 in fission yeast (**Samejima et al., 2010**), and ApsB in *Aspergillus nidulans* (**Zekert et al., 2010**), have a MASC motif and function at the cytoplasmic side of the SPB or in the cytoplasm. In case of metazoa, CM1-only type of γ-TuCRs such as CDK5RAP2 (**Wang et al., 2010**) possess a CM2 domain (**Figure 9B**, **Figure 8—figure supplement 1A**), targeting γ-TuCRs to both centrosomal and acentrosomal MTOCs.

In summary, γ-TuCR family proteins can be classified regarding the γ-TuSC interaction motifs and the MTOC targeting domain into three groups.

## Discussion

### Spc110 is a γ-TuSC-mediated nucleation regulator

In this study we show that the γ-TuCR Spc110 actively participates in MT formation through the SPM and CM1 elements and Cdk1 and Mps1 phosphorylation sites. Binding of γ-TuSC to Spc110 is a prerequisite for γ-TuSC oligomerization into the ring-like template that initiates MT assembly. Phosphorylation of Spc110 by Cdk1 and Mps1 regulates γ-TuSC binding and thereby MT nucleation. Thus, Spc110 is a MT nucleation regulator. We thereby provide a molecular understanding of the previously observed spindle length phenotypes (the two Cdk1-Clb5 sites) and genetic interaction with *SPC97* (three Mps1 sites combined with S36) of *SPC110* phosphorylation site mutants (**Friedman et al., 2001**; **Huisman et al., 2007**). Our data also explain why an N-terminal fragment of Spc110 influences oligomerization of γ-TuSC when co-expressed in insect cells (**Kollman et al., 2010**).

SPBs duplicate in G1/S phase of the cell cycle in a conservative manner (**Byers and Goetsch, 1975**; **Adams and Kilmartin, 2000**; **Pereira et al., 2001**; **Figure 10A**). In late G1, a SPB precursor, named the duplication plaque, assembles on the cytoplasmic side of the nuclear envelope at a specialized substructure of the mother SPB, named bridge (**Byers and Goetsch, 1975**; **Jaspersen and Winey, 2004**). In G1/S the duplication plaque then becomes inserted into the nuclear envelope. This allows binding of the Spc110-calmodulin-Spc29 complex from the nuclear side to the central Spc29-Spc42 layer of the embedded duplication plaque (**Bullitt et al., 1997**; **Elliott et al., 1999**). In this cell cycle phase Mps1 and Cdk1-Clb5 phosphorylate the N-terminus of Spc110 at five sites to promote γ-TuSC binding to Spc110 (**Figure 10B**). Consistently, Cdk1-Clb5 associates with SPBs throughout this time window (**Huisman et al., 2007**). These phosphorylations increase γ-TuSC affinity for Spc110-5P and induce γ-TuSC oligomerization into ring-like complexes that promote MT nucleation (**Figure 10B**). Phospho-regulation of Spc110 is not absolutely essential for viability as indicated by the growth of *spc110^{5A}* cells. However, *spc110^{5A}* cells have growth defects in the absence of the SAC gene *MAD2* and fail to organize the full set of MTs in S phase and mitosis (**Figure 6**). We therefore suggest that Mps1 and Cdk1-Clb5 coordinate the timing of MT nucleation through Spc110 phosphorylation with SPB duplication. A SAC induced cell cycle delay can temper any defects in this phospho-regulation as indicated by the genetic interaction between *spc110^{2A}* and *spc110^{5A}* with *mad2Δ* (**Figure 6A**).

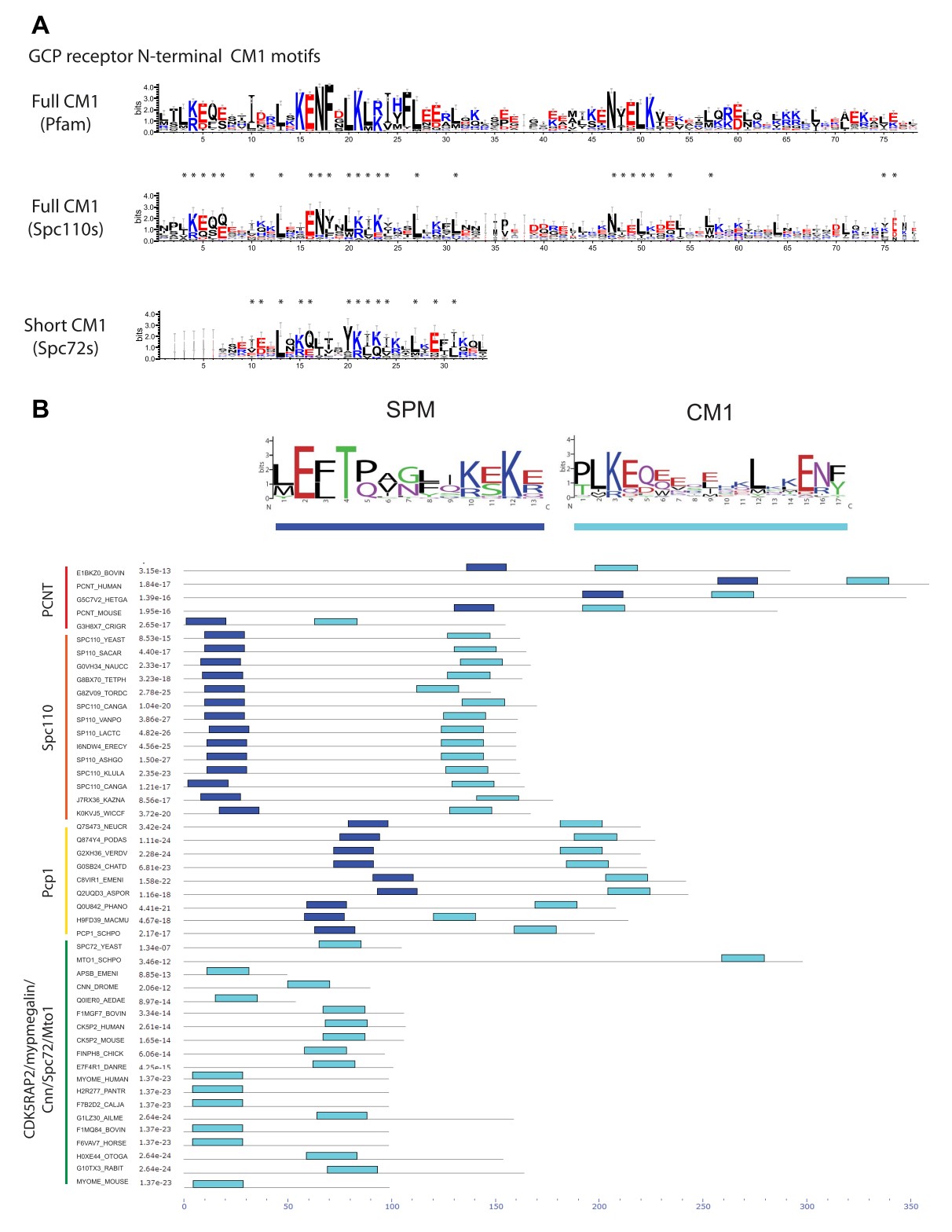

**Figure 8**. Two types of γ-TuCRs defined by the N-terminal γ-TuSC binding motifs: SPM-CM1 and CM1-only. (**A**) Graphical representations of the patterns of CM1 motif within the multiple sequence alignment of γ-TuCR protein sequences. The CM1 motif sequence logos were shown for γ-TuCR protein sequences retrieved from Pfam database (Microtub_assoc, Pfam id: PF07989), Spc110s and Spc72s from subphylum Saccharomycotina. The Spc72 sequences used to generate the short CM1 logo are shown in **Figure 8—figure supplement 1C**. The sequence logos were generated with Weblogo
*Figure 8. Continued on next page*

*Figure 8. Continued*

3.0 server. The conserved residues shared by CM1 defined by Pfam, Spc110s, and Spc72s are marked with asterisks. (**B**) SPM and CM1 in PCNT, Spc110, and Pcp1 homologues (SPM-CM1 type) and in CDK5RAP2/Cnn/Spc72/Mto1 (CM1 type). The occurrences of the motif in the training set sequences were calculated and shown as coloured blocks on a line with MEME motif scanning analysis (*Bailey et al., 2009*). Weblogo motif diagram is shown for each sequence showing the sites contributing to that motif in that sequence. The best p-value for the sequence/motif combination is listed. The p-value of an occurrence is the probability of a single random subsequence with the length of the motif, generated according to the zero-order background model, having a score at least as high as the score of the occurrence.

The following figure supplements are available for figure 8:

**Figure supplement 1**. Raw sequence input for the sequence logo of CM2, Spc110 PACT, and Spc72 CM1 motifs.

Spc110$^{T18}$ is an additional Cdk1 in vivo site (*Albuquerque et al., 2008*; *Keck et al., 2011*; *Lin et al., 2011*; *Figure 5—figure supplement 1*). Analysis of T18A, T18V, T18E, and T18D mutations resulted in similar in vitro and in vivo phenotypes. We have to assume that all these mutations inactivate the important SPM motif in which T18 resides. The fact that T18 of Spc110 is phosphorylated only by Cdk1-Clb2 and not by Cdk1-Clb5 and the in vivo phosphorylation of SPB-associated Spc110$^{T18}$ in mitosis but not in S phase (*Keck et al., 2011*; *Figure 5—figure supplement 1A-C*) indicates that this Cdk1 site is different compared to the two S phase Cdk1 sites S36 and S91 in Spc110. At least in vitro, Cdk1-Clb2 phosphorylation of Spc110$^{T18}$ inactivates in a dominant manner the γ-TuSC oligomerization activity (*Figure 5—figure supplement 1D,E*). Thus, it is well possible that Spc110$^{T18}$ phosphorylation by Cdk1-Clb2 limits the MT nucleation activity of the SPB associated γ-TuSC. Most likely only the surplus of γ-TuSC that is not engaged in MT organization is phosphorylated and affected by Spc110$^{pT18}$. This model fits with the observation that the yeast SPB of haploid cells organizes a constant number of 20 nuclear MTs during mitosis (*Winey et al., 1995*).

Co-overexpression of *SPC97*, *SPC98*, and *TUB4* does not induce MT nucleation despite of γ-TuSC assembly (*Pereira et al., 1998*). Our and other studies have shown that the missing factor promoting γ-TuSC ring assembly is Spc110 (*Vinh et al., 2002*; *Kollman et al., 2010*). In vivo data indicate that the binding of γ-TuSC to Spc110, although a prerequisite, is insufficient for MT formation. For example, although the γ-TuSC component Spc97 bound to mitotic SPBs of *spc110$^{T18D}$* or *spc110$^{5D-T18D}$* cells with similar efficiency to wild type cells, the γ-TuSC in the two mutants was insufficient in full MT organization (*Figure 6B*, *Figure 6—figure supplement 1C*). Consistent with this model, the addition of Spc110$^{1–220-5D-T18D}$ to recombinant γ-TuSC induced the formation of γ-TuSC dimers at ~600 kDa (*Figure 2—figure supplement 2A*; note that the protein concentration in this experiment was two-times higher than in *Figure 2—figure supplement 2B*) without the formation of γ-TuSC oligomers. We therefore suggest that γ-TuSC binding to Spc110 and TuSC oligomerization are mechanistically distinct steps (*Figure 10C*).

## The N-terminus of Spc98 mediates interaction with N-Spc110 and oligomerization

The N-terminal region of Spc98 exhibits homology to other GCP3 family members including human GCP3 (hGCP3, *Figure 7A*, *Figure 7—figure supplement 1A*). All homologues contain five predicted helical regions that are followed by an unstructured linker region before the two GRIP domains that are common to all GCP proteins (*Guillet et al., 2011*). Analysis of the structure of yeast γ-TuSC by electron microscopy localised N-Spc98 at of the base of the Y shaped structure, away from the C-Spc98 that interacts with γ-tubulin. N-Spc98 is close to the N-terminus of Spc97 and N-Spc110 (*Choy et al., 2009*; *Kollman et al., 2010*). This is consistent with yeast two-hybrid studies that identified N-Spc98 as an interactor for Spc110 (*Knop and Schiebel, 1997*; *Nguyen et al., 1998*). We have analyzed the importance of N-Spc98 by deleting different portions of this domain including the helices, the linker region, and both the helices and the linker. Surprisingly, the N-Spc98 fragment was not essential for the viability at 23°C (*Figure 7B*, *Figure 7—figure supplement 1B*). The important function of this region, however, was revealed at elevated temperatures and when SAC function was impaired (*Figure 7B*).

Biochemical analysis of γ-TuSC$^{Spc98Δ1–156}$ showed that its binding to Spc110$^{1–220-5D}$ and oligomerization capability by Spc110$^{1–220-5D}$ were reduced (*Figure 7E–G*), although in M phase it bound with

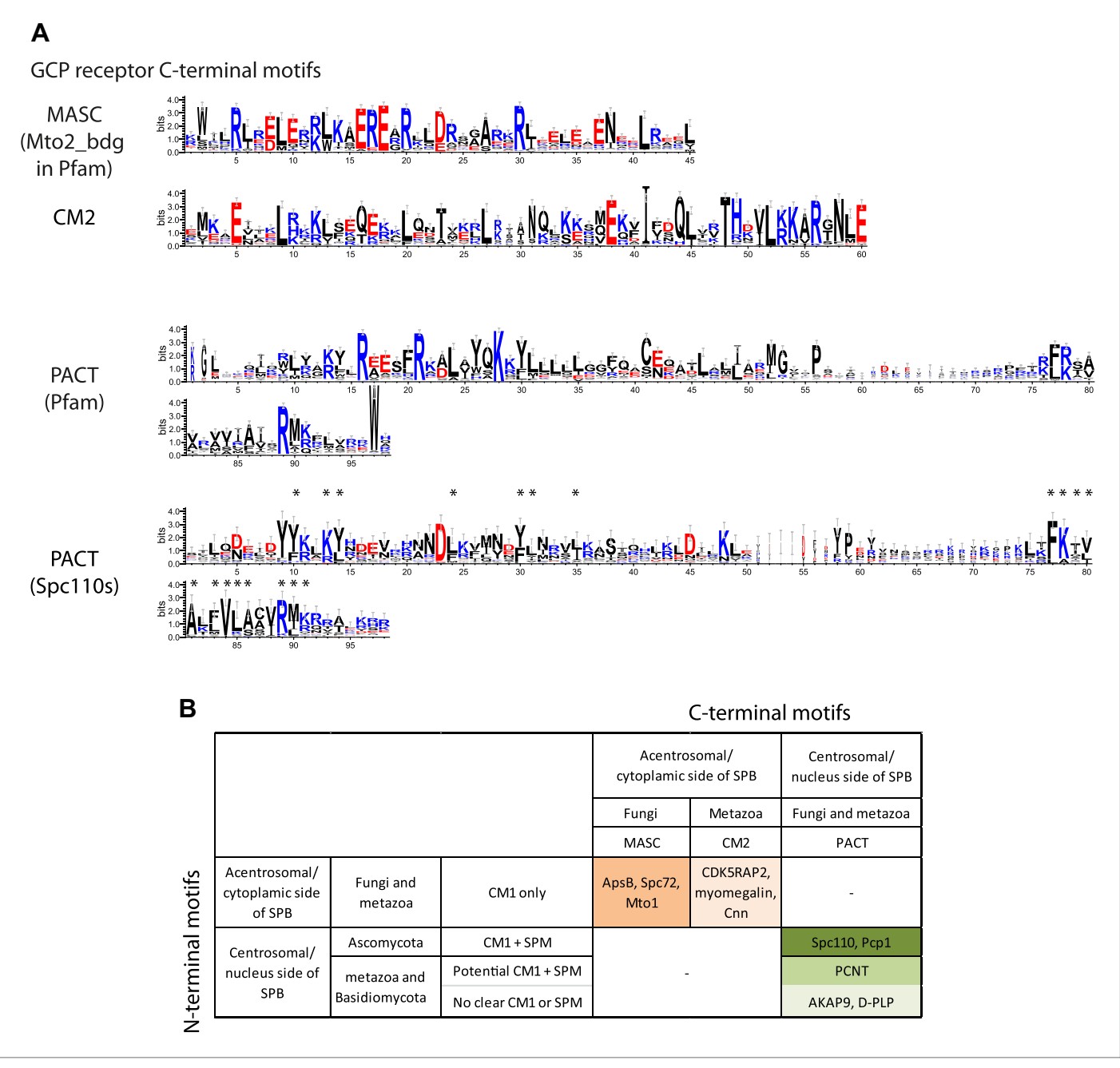

**Figure 9**. γ-TuCRs are classified into three subgroups based on N-terminal γ-TuSC binding motifs and C-terminal MTOC targeting motifs. (**A**) Graphical representations of the patterns of C-terminal MTOC targeting motifs within the multiple sequence alignment of γ-TuCR protein sequences. The MASC and PACT motif sequence logos were shown for γ-TuCR protein sequences retrieved from Pfam database (Mto2_bdg, Pfam id: PF12808; PACT_coil_coil, Pfam id: PF10495) and Spc110s from subphylum Saccharomycotina. The γ-TuCR and Spc110 protein sequences used to generate the CM2 and PACT motif logo are shown in *Figure 8—figure supplement 1A,B*. The sequence logos were generated with Weblogo 3.0 server. The conserved residues shared by PACT motif defined by Pfam and Spc110s are marked with asterisks. (**B**) Summary table of categorization of γ-TuCR protein family. Based on the presence of N-terminal γ-TuSC interacting motifs (CM1 and SPM) and C-terminal MTOC targeting motifs (MASC, CM2, and PACT), γ-TuCR protein family can be divided into three subgroups: N-terminal CM1 motif only with either C-terminal MASC or CM2 motifs, and N-terminal CM1 and SPM motifs combined with C-terminal PACT domain. Some PACT domain proteins (AKAP9 or D-PLP) have yet undefined γ-TuSC binding motifs.

the same affinity for SPBs as WT γ-TuSC (*Figure 7D*). We suggest that the special SPB arrangement of Spc110 as hexameric units (*Muller et al., 2005*) compensates in part for the reduced in vitro binding of γ-TuSC$^{Spc98\Delta1–156}$ to Spc110$^{1–220}$. This compensation may arise from cooperative interactions of γ-TuSC

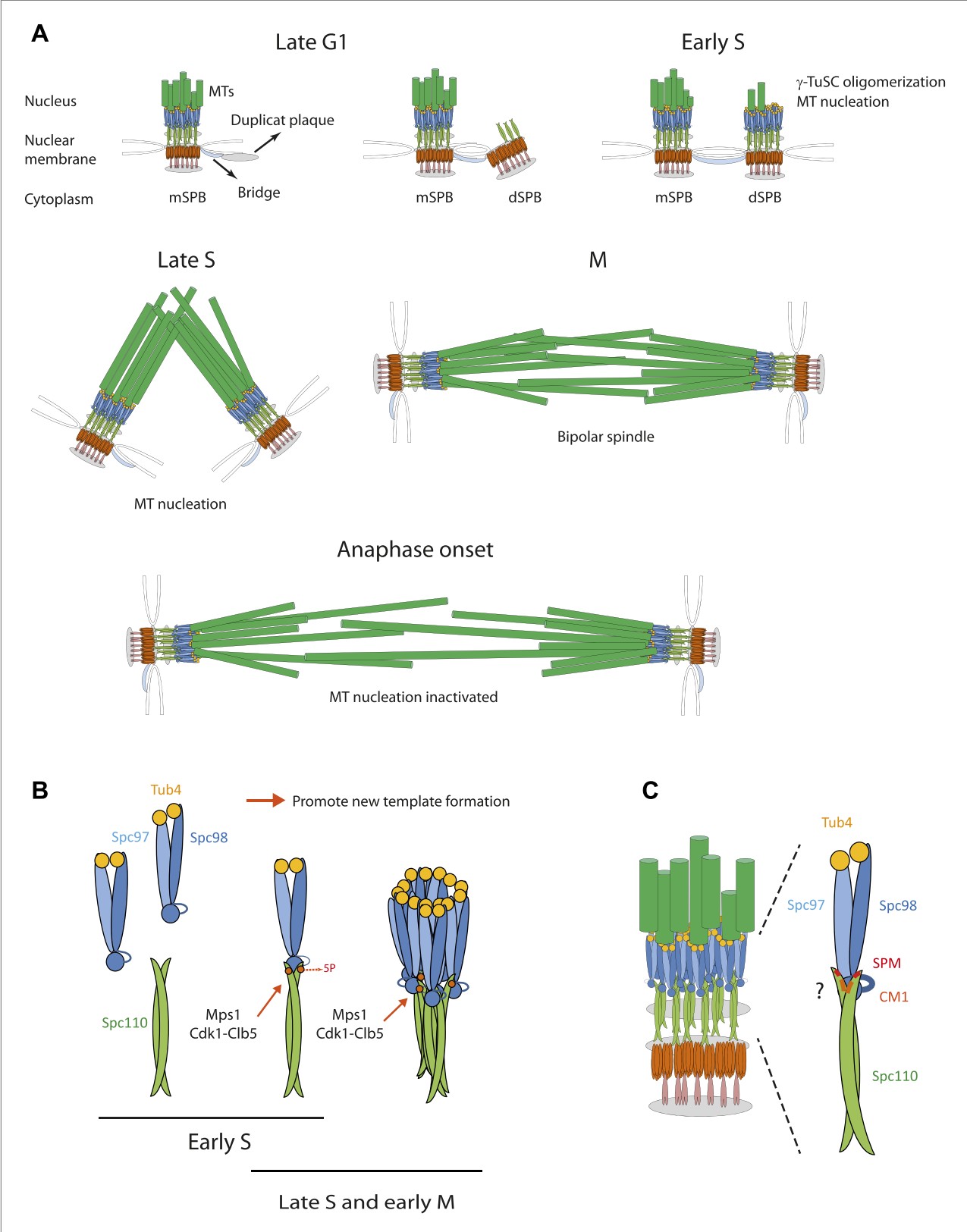

**Figure 10**. Role of Spc110 phosphorylation during SPB duplication. (**A**) MT nucleation by the SPB during the cell cycle. See 'Discussion' for details. (**B**) Cell cycle dependent, stimulatory phosphorylations of Spc110 by Mps1 and Cdk1-Clb5 (early S phase to early M). (**C**) Model for the interaction of γ-TuSC with Spc110. The Spc110 dimer interacts via the SPM and CM1 motifs with the N-terminus of Spc98 (GCP3) and possibly also with other regions of γ-TuSC.

with N-Spc110. Whatever the mechanism, regions of γ-TuSC in addition to the N-terminal 177 amino acids of Spc98 (*Figure 7—figure supplement 1B*) have to interact with Spc110. Either Spc110 also interacts with the Spc98/GCP3 core or it binds to the N-terminus of Spc97/GCP2.

γ-TuSC^Spc98Δ1–156 was still able to oligomerize under special buffer conditions (*Figure 7—figure supplement 2*), which is in agreement with the observation that γ-TuSC^Spc98Δ1–156 was functional in vivo and promoted MT nucleation, although defects were seen at elevated temperature and in *mad2Δ* cells. Thus, the regions that are essential for γ-TuSC oligomerization await identification.

## γ-tubulin complex receptors fall into three groups: the SPM-CM1-PACT, CM1-MASC, and CM1-CM2 types

γ-TuCRs from yeast to *Drosophila* to human carry a conserved CM1 motif. This CM1 motif was first identified in fission yeast Mto1 (*Sawin et al., 2004*). Subsequently the function of CM1 in γ-TuSC binding was confirmed for Mto1, centrosomin (*Drosophila*), and CDK5RAP2 (human) (*Zhang and Megraw, 2007*; *Fong et al., 2008*; *Samejima et al., 2008*; *Choi et al., 2010*). The γ-TuCRs Pcp1 (fission yeast) and Spc110 (budding yeast) also contain putative CM1 motifs (*Dictenberg et al., 1998*; *Takahashi et al., 2002*; *Zimmerman et al., 2004*; *Fong et al., 2010*; *Figure 8B*).

In addition, we identified a yet unappreciated CM1 motif in the yeast γ-TuCR Spc72 (*Knop and Schiebel, 1998*; *Figure 8A*). Although the putative CM1 motif in Spc72 appears degenerated compared with the full CM1 of other γ-TuCRs, the conserved residues shared by Spc72-CM1 and full CM1 suggest that Spc72 contains a functional CM1. Consistently, Spc72 carries a C-terminal MASC, as is the case for other fungal CM1-only γ-TuCRs (*Samejima et al., 2010*). Moreover, the predicted CM1 resides in the N-terminal region of Spc72 that is essential for γ-TuSC binding (*Knop and Schiebel, 1998*; *Usui et al., 2003*).

Human PCNT falls in terms of structure and function into the γ-TuCR class (*Doxsey et al., 1994*; *Dictenberg et al., 1998*; *Gillingham and Munro, 2000*). Analysis of the PCNT sequence for CM1 in combination with the newly identified SPM identified a potential CM1 in the N-terminus of PCNT (*Figure 8B*). This is consistent with the γ-tubulin binding ability of human PCNT, which lies within the region that contains the SPM-CM1 (*Takahashi et al., 2002*). In addition, super-resolution analysis of human centrosomes has mapped the localization of γ-tubulin towards the N-terminus of PCNT (*Lawo et al., 2012*; *Mennella et al., 2012*; *Sonnen et al., 2012*). Experimental evidences are awaited to validate the CM1 motif in Spc72 and PCNT.

Our study has identified SPM as a second conserved motif amongst a subgroup of γ-tubulin complex receptors (*Figure 2H*), namely the yeast Spc110 homologues, fission yeast Pcp1, and human PCNT (*Figure 10*; *Doxsey et al., 1994*; *Dictenberg et al., 1998*; *Flory et al., 2002*). The distance between SPM and CM1 is around 90–100 amino acids in the Spc110 and Pcp1 type of receptors. In the mammalian PCNT subfamily it is however only around 70 amino acids. Whether these differences are important for regulation or are indicative of co-adaptation with alterations in the γ-TuSC binding sites remains to be seen.

The SPM is important for Spc110 binding to γ-TuSC and induced oligomerization of γ-TuSC (*Figures 3 and 4*). Based on these results we propose that SPM co-operates with CM1 to form a bipartite binding element for the γ-TuSC (*Figure 10C*). Interestingly, however, other γ-tubulin complex receptors such as Spc72, fission yeast Mto1, *Drosophila* Cnn, and human CDK5RAP2 carry only the CM1 element but no SPM motif (*Figure 8*). Strikingly, SPM-CM1 type of γ-TuCRs carry a C-terminal PACT domain that targets these proteins to MTOCs (*Figure 8B*). The CM1-only type of γ-TuCRs either contain a MASC (fungi) or a CM2 C-terminal MTOC targeting sequence (metazoa) (*Figure 9B*). We also noticed that in some metazoa and the fungi phylum Basidiomycota the PACT domain-containing γ-TuCRs (such as AKAP9 in human and D-PLP in *Drosophila*) have N-terminal regions distinctive from SPM and CM1 motifs, probably reflecting different binding partners in γ-TuRC. Altogether, we can classify γ-TuCRs in at least three classes: CM1-MASC, CM1-CM2, and SPM-CM1-PACT.

In budding and fission yeast, the CM1-MASC types of γ-TuCRs (Spc72 and Mto1) have specific functions in cytoplasmic MT organization, while Spc110 and Pcp1 (SPM-CM1-PACT) organize only nuclear MTs. This functional division is accompanied by the physical separation of the receptors by the nuclear envelope, which remains intact during the closed mitosis of both organisms. Human CDK5RAP2, myomegalin (CM1–CM2 types), and pericentrin (SPM-CM1-PACT type) are associated with the centrosome (*Doxsey et al., 1994*; *Fong et al., 2008*). However, CDK5RAP2 and myomegalin also have

functions in acentrosomal MT nucleation for example MT nucleation by the Golgi (*Choi et al., 2010*; *Roubin et al., 2013*). Thus, it may be the acentrosomal/cytoplasmic function that explains the lack of SPM element in Mto1/CDK5RAP2/myomegalin.

Budding yeast organizes MTs only with γ-TuSC and γ-TuCRs. Other organisms have additional building blocks that contribute to MT assembly, such as Mozart1 and GCP4-6 (*Dhani et al., 2013*; *Masuda et al., 2013*). However, GCP4 to 6 are not essential for MT nucleation in fission yeast and *Drosophila*, while GCP2/Alp4, GCP3/Alp6, and Mozart1 are essential for viability (*Anders et al., 2006*; *Verollet et al., 2006*). Puzzlingly, fission yeast Mozart1 is essential for γ-TuSC recruitment to SPBs (*Masuda et al., 2013*). This raises the question how budding yeast compensates for Mozart1's function. GCP3 and γ-TuCRs of the SPM-CM1 type are conserved between organisms with or without Mozart1. It is likely that small adaptive changes in GCP3 or Spc110 have evolved to compensate for the lack of Mozart1 in budding yeast.

What could be the function of γ-TuCR in organisms that are able to assemble the more stable γ-TuRC? γ-TuCRs may recruit and activate the already assembled γ-TuRC to MTOCs via GCP interactions. In addition, γ-TuCRs could induce γ-TuSC assembly into rings that then either function without the help of additional GCPs or subsequently become stabilized by GCP4-6. In any case, considering the conservation between the SPM and CM1 binding elements of γ-TuRCs and the N-terminal region of Spc98/GCP3, we suggest that the basic principals of MT nucleation are conserved from yeast to human.

## Materials and methods

### Plasmid and strain constructions

A detailed list of DNA constructs and yeast strains is described in *Supplementary file 1* and *Supplementary file 2*. *SPC110* alleles were subcloned into the integration vector pRS304, and *SPC98* alleles were subcloned into pRS305K (*Sikorski and Hieter, 1989*; *Christianson et al., 1992*; *Taxis and Knop, 2006*). Point mutations in genes were introduced by PCR-directed mutagenesis and confirmed by DNA sequencing. GST-Spc110$^{1–220}$ was cloned into pGEX-5X-1 vector (GE Healthcare, UK) and His-tagged Mps1 was cloned into pET28b for expression in BL21 CodonPlus *E. coli* and into pFastBac1 vector for overexpression in baculovirus-insect cell system. All yeast strains are derivatives of S288c. Gene deletion and epitope tagging of genes at their endogenous loci performed using standard techniques (*Knop et al., 1999*; *Janke et al., 2004*). The red fluorescent mCherry was used to mark SPBs through a fusion with *SPC42* (*Donaldson and Kilmartin, 1996*). For *SPC97-yeGFP* and *SPC110-yeGFP* strains, the endogenous *SPC97* and the integrated *SPC110* alleles were tagged with green fluorescent yeGFP. *GFP-TUB1* strains were constructed using an integration plasmid (*Straight et al., 1997*). To remove *URA3*-based plasmids, transformants were tested for growth on 5-fluoroorotic acid (5-FOA) plates that select against *URA3*-based plasmids.

### Antibodies

Affinity-purified anti-Tub4 (1:1000) and anti-Spc110 (1:1000) antibodies were described previously (*Geissler et al., 1996*; *Spang et al., 1996a*, *1996b*). Anti-His-tag (1:1000) and anti-GST (1:1000) antibodies were used to detect Spc97 and Spc98 of recombinant γ-TuSC. Secondary antibodies used in semi-quantitative blotting were IRDye800- or Alexa680-conjugated anti-goat, anti-rabbit, anti-guinea pig, and anti-mouse IgGs (1:50,000; Rockland Immunochemicals Inc., Gilbertsville, PA). Phospho-specific anti-Spc110$^{pT18}$, anti-Spc110$^{pS36-pS91}$, and anti-Spc110$^{pS60-pT64-pT68}$ antibodies were raised in guinea pigs and purified with immuno-affinity chromatography (1:200; Peptide Specialty Laboratories GmbH, Germany).

### Protein purification

Subunits of γ-TuSC and γ-TuSC$^{ΔN-Spc98}$ were expressed with the baculovirus-insect cell expression system. Purification was performed as described (*Gombos et al., 2013*). Spc110$^{1-220}$ variants with an amino-terminal GST tag were purified with Glutathione Sepharose 4B (GE Healthcare, UK) as described (*Vinh et al., 2002*). Proteins were eluted with 5 mM glutathione, concentrated, and run over a Superose 6 column equilibrated in HB100 to remove glutathione. Recombinant Mps1 protein was purified in lysis buffer (50 mM NaH$_2$PO$_4$ pH 7.5, 300 mM NaCl, 10% glycerol, 1 mM DTT, 1 mM PMSF, 1X Complete EDTA-free protease inhibitor cocktail (Roche, Canada)) with column packed with Ni-NTA Sepharose (GE Healthcare, UK). After wash with lysis buffer, protein was eluted with lysis buffer containing

200 mM imidazole. Small aliquots for kinase assay were snap frozen with liquid nitrogen and stored at −80°C. Cdk1$^{as1}$-Clb2 and Cdk1$^{as1}$-Clb5 complexes were purified from yeast strains kindly given by David Morgan (see *Supplementary file 2* for detail) (*Loog and Morgan, 2005*). *CLB2-TAP* and *CLB5-TAP* were encoded on 2-micron plasmid and driven by Gal1 promoter. After overexpression with 2% galactose, cells were harvested and lysed for TAP-tag purification as previously described (*Ubersax et al., 2003*).

## γ-TuSC oligomerization assay

Purified γ-TuSC and γ-TuSC$^{Spc98ΔN}$ were mixed with or without Spc110$^{1–220}$ in 1:4 molar ratio (2.3:9.3) in TB150 (50 mM Tris–HCl, pH7.0 with 150 mM KCl) or BRB80 (80 mM PIPES pH6.9, 1 mM EGTA, 1 mM MgCl$_2$) buffer as indicated in figure legends. After incubating for 1 hr at 4°C, the protein mixture was applied to a gel filtration column (Superdex 200 10/300 or Superose 6 10/300 [GE Healthcare, UK]). Elution profiles were recorded as absorbance at 280 nm. For each fraction 2% of sample volume was analyzed by SDS-PAGE. Proteins were detected by silver staining or immunoblotting.

## In vitro binding assay

For the measurement of the binding of γ-TuSC to Spc110$^{1–220}$, recombinant GST (300 nM) or GST-Spc110$^{1–220}$ proteins (from 0 to 300 nM or fixed 300 nM) were incubated with γ-TuSC (150 nM) in TB150 buffer in a total reaction volume of 30 µl on a rocking platform for 0.5 hr at 4°C. Glutathione-Sepharose 4B bead slurry (20 µl; GE Healthcare, UK) was then added to each reaction, followed by rocking for an additional 1 hr at 4°C. Beads were washed five times with TB150 buffer, 0.1% NP40 followed by heating in 20 µl SDS sample buffer. Input and bound proteins were analysed by immunoblotting. The signal intensities of protein bands on immunoblots were quantified with ImageJ (NIH). Signal intensities were corrected against the membrane background.

## In vitro MT nucleation assay

Microtubule nucleation assay was performed as described (*Gombos et al., 2013*) with some modifications. 3 µM γ-TuSC was pre-incubated with 15 µM GST-Spc110$^{1–220}$ in HB100 buffer plus 12.5% glycerol for 30 min on ice. An equal volume of 20 µM bovine brain tubulin containing 4% Alexa568-labelled tubulin in BRB80 buffer with 25% glycerol was added and samples were further incubated for 30 min on ice before being transferred to 37°C for 15 min for MT polymerization. Samples were fixed with 1% formaldehyde and diluted with 1 ml cold BRB80 buffer. 50 µl aliquot of reaction mixture was sedimented onto poly-lysine-coated coverslips with a HB-4 rotor at 25,600 g for 1 hr. Samples were post-fixed with pre-cooled methanol, mounted on slides with CitiFluor mounting media, and imaged with a fluorescence microscope as described. Microtubules were counted in 20 random fields.

## In vitro kinase assay

For the in vitro phosphorylation of Spc110$^{1–220}$ variants, 0.5 µg Spc110$^{1–220}$ substrate was incubated in kinase buffer (50 mM Tris–HCl pH 7.5, 10 mM MgCl$_2$, 2 mM EGTA) with purified Mps1 kinase or Cdk1$^{as1}$ kinase co-immunoprecipitated with Clb2-TAP or Clb5-TAP. For autoradiography, 5 µCi [γ-$^{32}$P]-ATP was added to the reaction mixture and the kinase reaction was performed for 30 min at 30°C. The reaction was stopped by the addition of SDS sample buffer and analysed by SDS–PAGE, Simply Safe Blue staining (Invitrogen, Carlsbad, CA), and autoradiography. For phospho-specific antibody detection, 0.5 mM unlabeled ATP was used in the kinase reaction. For the γ-TuSC oligomerization assay by gel filtration, the phosphorylation reaction was stopped by addition of 15 µM 1NM-PP1 to inhibit Cdk1$^{as1}$ or 10 µM SP600125 (Sigma-Aldrich, St Louis, MO) to inhibit Mps1 kinase.

## Electron microscopic analysis of γ-TuSC oligomers

To visualize γ-TuSC monomers and oligomers, proteins were stained with uranyl acetate and analyzed by electron microscopy. Briefly, we evaporated 300 or 400 mesh copper/palladium grids with a carbon layer on the copper side. To enhance the hydrophilic affinity of the carbon layer, grids were glow discharged for 30–45 s directly before the preparation started. The protein solution was incubated for 30 s on the carbon mesh grids at room temperature. The grids were washed and incubated in 2% uranyl acetate for 4 min and blotted on Whatman 50 paper. The images were taken in low dose modus with an under-focus between 0.8 and 1.5 µm. Particles were viewed with a CM120 electron microscope (Philips Electronics NV, Eindhoven, Netherlands), which was operated

at 120 kV. Images were captured with a CCD camera (Keen View, Soft Imaging systems, Germany) and viewed with Digital Micrograph Software.

## Phosphopeptide enrichment and mass spectrometry

Phosphopeptides were identified according to Villen et al. (*Dephoure et al., 2008*). In brief, SDS-PAGE separated Spc110 was cut out and subjected to trypsin digestion. After digest, phosphopeptides were enriched by IMAC, starting with 25 µl of PHOS-Select beads (Sigma, Germany). Enriched phosphopeptides were eluted and desalted by C18 columns (Stage tips). Peptides were analyzed by LC-MS/MS (Orbitrap Elite, Thermo Fisher Scientific, Waltham, MA) and data were processed using Thermo proteome discoverer software (1.4). Phosphorylation site localization was performed on the Mascot results using PhosphoRS.

## Growth assay

Yeast cells in the early log phase were adjusted to an $OD_{600}$ of 1 with PBS. 10-fold serial dilutions of cells were spotted onto the indicated plates and incubated as indicated in the figure legends.

## Quantification of the Spc97-GFP, Spc110-GFP, and GFP-Tub1 signal at SPBs

Asynchronous cells were grown in filter-sterilized YPD with additional 0.1 mg/l adenine (YPAD) to an $OD_{600}$ of 0.3 at 23°C for 3 hr and then shifted to 37°C for 1 hr. Cells were directly sampled for image acquisition. For the image acquisition, z-stack images with 21 0.3 µm steps (2 × 2 binning) were acquired at 37°C with a DeltaVision RT system (Applied Precision, UK) equipped with FITC (fluorescein isothiocyanate), TRITC (tetramethyl rhodamine isothiocyanate), and Cy5 filters (Chroma Technology, Bellows Falls, VT), a 100x NA 1.4 plan Apo oil immersion objective (IX70; Olympus, Japan), and a CCD camera (CoolSNAP HQ; Roper Scientific, Tucson, AZ). Images were processed and analyzed in ImageJ (NIH).

The quantification of the mean background intensity and mean fluorescence intensity of Spc97-GFP and Spc110-GFP signals at SPBs was performed on planes having SPBs in focus. Spc97-GFP or Spc110-GFP intensity within the 3 × 3 pixel-area covering a single SPB or two unsplit SPBs was measured. For GFP-Tub1 at SPBs, GFP-intensity within a 3 × 3 pixel-area surrounding the SPB was measured. Unsplit SPBs were measured together, whereas split SPBs were measured separately. The standard error (SEM) for each data set (n = 50) and level of significance were determined by one-way ANOVA with Turkey's multiple comparisons' test.

## Time-lapsed live cell imaging

Cells were synchronized with α-factor for 1.5 cell cycles and then immobilized onto glass-bottom dishes. Dishes were prepared by incubating them with 100 µl concanavalin A solution (6% concanavalin A, 100 mM Tris-Cl, pH 7.0, and 100 mM $MnCl_2$) for 5 min and subsequently washed with 300 µl of distilled water. Yeast cells were allowed to attach to the dishes for 5–15 min at 30°C. The α-factor blocked cells were then released by two washes with 2 ml of prewarmed SC medium. Image acquisition was started 10–15 min after release from α-factor. Conditions for imaging were as follows: 15 stacks in the FITC channel, 0.1-s exposure, 0.3-µm stack distance, one reference image in bright field channel with a 0.05-s exposure, and 61 frames in total every 3 min. Images were quantified by measuring the integrated density of the sum of projected videos for the region of interest (ROI) around the SPBs and a background region selected from the periphery of the analyzed regions. The mean intensity of the background was subtracted from the ROI. To correct for acquisition, bleaching signal intensities were divided by a bleaching factor. The bleaching factor was determined from the mean of three very short videos that had been generated with the same image acquisition conditions. The data points of bleaching factor were fitted with non-linear one-phase decay equation.

## Nocodazole washout assay (MT regrow assay)

Cells (5 × 10^6 cells/ml) were pre-grown at 23°C in filter-sterilized YPAD. Cells were arrested in G1 by treatment with 10 µg/ml α-factor (Sigma-Aldrich, Germany) for 3 hr at 23°C until >95% of cells showed a mating projection. G1 arrested cells were released into media with hydroxyurea to arrest cells in S-phase for 2 hr. The culture was then shifted to 37°C and nocodazole was added. After 1 hr, nocodazole was removed by exchanging media with YPAD containing hydroxyurea. After removal of nocodazole, cells were sampled at 0, 30, and 60 min and fixed with 4% paraformaldehyde. Image acquisition and GFP-Tub1 quantification were performed as described in previous section.

### Trichloroacetic acid (TCA) extraction of yeast cells

To measure the expression level of Spc110 variants and GFP-Tub1 in vivo, whole cell lysates were prepared for SDS-PAGE and immunoblotting (*Knop et al., 1999*; *Janke et al., 2004*). 2–3 $OD_{600}$ of late-log phase liquid culture were resuspended in 0.2 M NaOH and incubated on ice for 10 min. 150 µl 55% (wt/vol) TCA was added and the solutions were mixed and incubated for 10 min on ice. After centrifugation the supernatant was removed. The protein pellet was resuspended in high urea (HU) buffer (8 M urea, 5% SDS, 200 mM Tris–HCl pH 6.8, 0.1 mM EDTA, 100 mM DTT, bromophenol blue) and heated at 65°C for 10 min. One-fifth of total sample amount was loaded for SDS-PAGE and western blotting.

### Pull-down of recombinant Spc110 protein

For in vivo phospho-Spc110 detection, strains with integrated *SPC110*-yeGFP or SPC110-TAP alleles were lysed in binding buffer (25 mM HEPES, pH 7.4, 150 mM NaCl, 1 mM EDTA, 1 mM DTT, 50 mM NaF, 80 mM β-glycerophosphate, 0.2 mM $Na_3VO_4$, and protease inhibitors) lysed with acid-washed glass beads (Sigma-Aldrich, Germany) in a FastPrep FP120 Cell Disrupter (Thermo Scientific, Germany). Cell lysates were incubated with 0.1% Triton X-100 for 15 min and then clarified by centrifugation at 10,000×*g* for 10 min. To further solubilize the pellet with insoluble SPBs, binding buffer containing 0.5 M NaCl and 1% Triton X-100 was added to resuspend the pellet and incubated for 40 min. GFP-binder protein (*Rothbauer et al., 2008*) conjugated Sepharose 4B slurry or IgG-conjugated Dynabeads was added into supernatants and rocked at 4°C for 2 hr, followed by five times of washing with binding buffer. Pull-downed Spc110-GFP or Spc110-TAP was eluted by heating with SDS sample buffer and analyzed by SDS-PAGE and immunoblotting or mass spectrometry.

### Bioinformatic analysis

Protein sequences of Spc110 and Spc98 and their homologues in selected organisms were aligned with MAFFT algorism built in Jalview software (*Waterhouse et al., 2009*; *Katoh and Standley, 2013*; *Figures 2H and 7A*, *Figure 2—figure supplement 6*, *Figure 8—figure supplement 1*). The threshold of conservation was set as 20% and highlighted. The relative frequency of each amino acid in each position of SPM and CM1 motifs was visualized using the WebLogo 2.0 (*Crooks et al., 2004*; *Figure 2H*, *Figure 8*, *Figure 9*, *Figure 2—figure supplement 6*, *Figure 8—figure supplement 1*). To demonstrate the occurrence of SPM and CM1 motifs in selected members of γ-TuSC receptor family, 47 protein sequences covering the putative CM1 motifs were subjected to MEME motif scanning analysis (*Bailey et al., 2009*; *Figure 8B*).

Jpred 3 and Disopred predicted the secondary structure of the N-terminal domain of Spc98 and human GCP3 (*Ward et al., 2004*; *Cole et al., 2008*). Domain positions and protein interacting regions of Spc110 and Spc98 (*Figure 1A*, *Figure 7—figure supplement 1A*) were illustrated according to γ-TuSC binding studies of Spc72- and Spc110-truncated forms and to the yeast-two-hybrid studies (*Sundberg and Davis, 1997*; *Knop and Schiebel, 1998*; *Usui et al., 2003*).

For the identification of full CM1 motif on Spc110s and degenerative CM1 on Spc72, CM1 motif containing γ-TuCR protein sequences were retrieved from Pfam database (Microtub_assoc, Pfam id: PF07989) and multiple-aligned with Spc110s and Spc72s from species of subphylum Saccharomycotina by MAFFT multiple sequence alignment algorism. To define the CM2 motif pattern, CM2 sequences from human CDK5RAP2 and *Drosophila* Cnn were used to retrieve protein sequences containing putative CM2 with HMMER server (*Finn et al., 2011*). The retrieved protein sequences were multiple-aligned with CDK5RAP2 and Cnn by MAFFT algorism and the multiple-aligned CM2 sequences were subjected to Weblogo 3.0 server to visualize the pattern.

### Statistical analysis

Statistical analysis of fluorescent intensities, immunoblotting intensities, and in vitro microtubule numbers was performed with GraphPad Prism 6.1. One-way ANOVA with Turkey's multiple comparisons' test was used to compare samples and to obtain adjusted p values. The number of repeated experiments and sample size are indicated in figure legends. No statistical method was used to predetermine sample size. The experiments were not randomized. The investigators were not blinded to allocation during the experiments and outcome assessment. In general the data showed normal distribution.

## Acknowledgements

This work was supported by a grant of the Deutsche Forschungsgemeinschaft (Schi295-3-2). Dr David Morgen is acknowledged for yeast strains. We thank U Jäkle and S Heinze for excellent technical

support. TL is a member of the international graduate school HBIGS and was supported by a fellowship of the Graduiertenkolleg Regulation of Cell Division.

## Additional information

### Funding

| Funder | Grant reference number | Author |
| --- | --- | --- |
| Deutsche Forschungsgemeinschaft | Schi295-3-2 | Tien-chen Lin, Annett Neuner, Yvonne T Schlosser, Elmar Schiebel |
| Deutsche Forschungsgemeinschaft (DFG) | SFB1036 | Annette ND Scharf, Lisa Weber |

The funders had no role in study design, data collection and interpretation, or the decision to submit the work for publication.

### Author contributions

TL, Conception and design, Acquisition of data, Analysis and interpretation of data, Drafting or revising the article; AN, Single particle analysis with electronic microscopy, Acquisition of data, Analysis and interpretation of data; YTS, Characterization of phenotypes of *spc98* mutant cells, Acquisition of data, Analysis and interpretation of data; ANDS, LW, MS analysis of phosphorylated peptides, Acquisition of data, Analysis and interpretation of data; ES, Conception and design, Drafting or revising the article

## Additional files

### Supplementary files

• Supplementary file 1. Plasmids used in this study.

• Supplementary file 2. Strains used in this study.

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
