## [Decision Letter]

Thank you for sending your work entitled “Cell-cycle dependent phosphorylation of yeast pericentrin regulates γ-TuSC-mediated microtubule nucleation” for consideration at *eLife*. Your article has been favorably evaluated by a Senior editor, a Reviewing editor, and 2 reviewers.

The Reviewing editor and the reviewers discussed their comments before we reached this decision, and the Reviewing editor has assembled the following comments to help you prepare a revised submission.

1) The primary concern is with T18. It is likely that this site is phosphorylated in vivo, but Spc110 mutant proteins containing T18D or T18A have the same rather than opposing properties. Therefore, the evidence that that phosphorylation of T18 is really inhibitory is lacking. (It is likely that T18D did not act as a phosphomimetic.) The strong effect of T18Ala is a concern. It is difficult to see why this should perturb the structure, as the authors claim. The authors could try a valine substitution instead.

2) It is puzzling that mitotic recruitment of Spc97 to the SPB of T18D mutants remains unchanged (Figure 7—figure supplement 1), although there is an effect on microtubule nucleation. This also contrasts with the effect of T18 phosphorylation on the oligomerization of gamma-TuSCs seen in vitro. It is important to resolve this discrepancy.

3) It would further strengthen this manuscript if the authors could demonstrate in a more refined way that the pT18 epitope peaks late in mitosis and is lost upon mitotic exit (e.g., by immunoblotting of synchronized cells with anti pT18 in a time course experiment). The anti-phospho-T18 blots in Figure 6 are not convincing.

4) The biological significance of T18 phosphorylation to inhibit microtubule nucleation is unclear. The dynamics of spindle microtubules are high during this mid-mitosis, thus to support this property it is likely that nucleation activities from the SPB should be high as well.

---

## [Author Response]

*1) The primary concern is with T18. It is likely that this site is phosphorylated in vivo, but Spc110 mutant proteins containing T18D or T18A have the same rather than opposing properties. Therefore, the evidence that that phosphorylation of T18 is really inhibitory is lacking. (It is likely that T18D did not act as a phosphomimetic.) The strong effect of T18Ala is a concern. It is difficult to see why this should perturb the structure, as the authors claim. The authors could try a valine substitution instead*.

As suggested by the reviewers, we have constructed *spc110-T18V* and in addition, *spc110-T18E* mutants. We have analyzed both mutants using biochemistry (Figure 2—figure supplement 5) and cell growth (Figure 2—figure supplement 5). The bottom line of both experiments is that *spc110-T18A* behaves as *spc110-T18V* and *spc110-T18D* behaves as *spc110-T18E*.

In addition, we have constructed a destruction box lacking *ΔDB-CLB2-SPC110* gene fusion following a strategy that was originally introduced by D. Morgan. We hoped that this gene fusion will target Cdk1 permanently to the SPB and phosphorylate T18 in a cell cycle independent manner. This should give us the *spc110-pT18* phenotype. However, the *CLB2-SPC110* gene fusion did not provide *SPC110* function (Figure 11.)Author response image 1.

To this end, we cannot say without ambiguities that T18D/E are phosphomimetic and T18A/V are phosphoinhibitory. However, it is clear that T18 lies within a conserved region, that we named SPM. Thus, we have changed the T18 phosphorylation part in this direction. In Figure 5—figure supplement 1 we now show that T18 of Spc110 is phosphorylated in mitosis and we also show the *in vitro* data on Spc110^1-220-5D-pT18^ on γ-TuSC oligomerization (Figure 5—figure supplement 1). These data are consistent with an inhibitory role of pT18. This role is now discussed in the Discussion.

*2) It is puzzling that mitotic recruitment of Spc97 to the SPB of T18D mutants remains unchanged (*Figure 7—figure supplement 1*), although there is an effect on microtubule nucleation. This also contrasts with the effect of T18 phosphorylation on the oligomerization of gamma-TuSCs seen in vitro. It is important to resolve this discrepancy*.

In response to this comment, we have performed the experiment shown in Figure 6—figure supplement 1. It shows that Spc97-GFP is recruited to SPBs of *spc110-T18D* cells with a time delay after SPB duplication. However, despite equal Spc97 amounts at SPBs in mitosis, *spc110-T18D* cells have a MT organization defect (Figure 6). Indeed, Figure 2—figure supplement 2 shows that Spc110^1-220-T18D^ and Spc110^1-220-5D-T18D^ form putative dimers with γ-TuSC (shoulder at 600 kDa) without inducing γ-TuSC oligomerization (Vo). This indicates that γ-TuSC recruitment to SPBs via Spc110 and γ-TuSC oligomerization into a nucleation platform are mechanistically two distinct steps. We discuss this scenario in the Discussion.

*3) It would further strengthen this manuscript if the authors could demonstrate in a more refined way that the pT18 epitope peaks late in mitosis and is lost upon mitotic exit (e.g., by immunoblotting of synchronized cells with anti pT18 in a time course experiment). The anti-phospho-T18 blots in*
Figure 6
*are not convincing*.

We tried to detect pT18 of Spc110 in IB with the P-specific anti-pT18 antibodies. However, this antibody only works with *in vitro* phosphorylated Spc110-pT18 (Figure 5—figure supplement 1). We therefore turned to mass spectrometry to determine the cell cycle dependent phosphorylation of Spc110. We have fractionated yeast cells into a supernatant fraction and a SPB containing pellet. We have extracted SPB proteins from the pellet. Spc110-GFP was enriched form the supernatant and the SPB faction. As shown in Figure 5—figure supplement 1 T18 of Spc110 is predominately phosphorylated in mitosis. This fits with the data from Keck et al., which showed that Spc110-T18 was phosphorylated in mitosis.

*4) The biological significance of T18 phosphorylation to inhibit microtubule nucleation is unclear. The dynamics of spindle microtubules are high during this mid-mitosis, thus to support this property it is likely that nucleation activities from the SPB should be high as well*.

As outlined under 1) we have changed this part of the paper. In budding yeast, nuclear MTs are only dynamic from the plus end because of the blocking cap of γ-TuSC at the MT minus end. Data from T. Stearns (Murphy et al., 1998) suggest that in budding yeast MT nucleation mainly happens in S phase when SPBs duplicate but probably not in other cell cycle phases. The T18 phosphorylation could inhibit excess MT nucleation by SPB-associated γ-TuSC in mitosis. SPBs organize relative constant numbers of 20 MTs in mitosis (87).

Because T18A is not behaving as a non-phosphorylatable mutation, we unfortunately cannot test this model. However, we have discussed in line 510 of the Discussion the possibility that pT18 restricts the number of nuclear MTs in mitosis.